# Osseosurface electronics—thin, wireless, battery-free and multimodal musculoskeletal biointerfaces

Le Cai[1,5], Alex Burton[1,5], David A. Gonzales[2], Kevin Albert Kasper[1], Amirhossein Azami [1], Roberto Peralta[3], Megan Johnson[1], Jakob A. Bakall[1], Efren Barron Villalobos[2], Ethan C. Ross[1], John A. Szivek[1,2], David S. Margolis [1,2,6✉] & Philipp Gutruf [1,4,6✉]

Bioelectronic interfaces have been extensively investigated in recent years and advances in technology derived from these tools, such as soft and ultrathin sensors, now offer the opportunity to interface with parts of the body that were largely unexplored due to the lack of suitable tools. The musculoskeletal system is an understudied area where these new technologies can result in advanced capabilities. Bones as a sensor and stimulation location offer tremendous advantages for chronic biointerfaces because devices can be permanently bonded and provide stable optical, electromagnetic, and mechanical impedance over the course of years. Here we introduce a new class of wireless battery-free devices, named osseosurface electronics, which feature soft mechanics, ultra-thin form factor and miniaturized multimodal biointerfaces comprised of sensors and optoelectronics directly adhered to the surface of the bone. Potential of this fully implanted device class is demonstrated via real-time recording of bone strain, millikelvin resolution thermography and delivery of optical stimulation in freely-moving small animal models. Battery-free device architecture, direct growth to the bone via surface engineered calcium phosphate ceramic particles, demonstration of operation in deep tissue in large animal models and readout with a smartphone highlight suitable characteristics for exploratory research and utility as a diagnostic and therapeutic platform.

[1] Department of Biomedical Engineering, University of Arizona, Tucson, AZ 85721, USA. [2] Department of Orthopaedic Surgery and Arizona Arthritis Center, University of Arizona, Tucson, AZ 85721, USA. [3] Department of Aerospace and Mechanical Engineering, University of Arizona, Tucson, AZ 85721, USA. [4] Departments of Electrical and Computer Engineering, BIO5 Institute, Neuroscience GIDP, University of Arizona, Tucson, AZ 85721, USA. [5] These authors contributed equally: Le Cai, Alex Burton. [6] These authors jointly supervised this work: David S. Margolis, Philipp Gutruf. ✉email: dsm@arizona.edu; pgutruf@email.arizona.edu

Continuous recording of bio-signals with high fidelity has been widely recognized to play a key role in modern exploratory research, diagnostics and therapeutics[1]. Specifically, with the emergence of computational tools such as neural networks, artificial intelligence, and machine learning that can help to analyze large datasets[2,3], continuous high-quality data streams will enable the development of diagnostics and therapeutics that will result in significantly improved patient outcomes[4,5]. However, current biosensing platforms with clinically relevant data streams rarely extend recording beyond short time periods. This is due to inadequate powering solutions, such as bulky electrochemical power supplies[6] and biointerfaces that degrade rapidly requiring intervention by users or health care providers, thus limiting the utility for exploratory, screening, diagnostic, and therapeutic applications[7,8].

Recent integration of high-performance silicon-based devices[9] and emerging soft electronics[10] yields numerous subdermally implantable neuromodulation systems that interface intimately with the central and peripheral nervous system[11,12]. This technological platform enables wireless supply of power and communication for highly miniaturized implants, allowing for the creation of interfaces to organs that are currently understudied due to the lack of suitable tools. One such area is the musculoskeletal system where wireless, battery-free interfaces are critical to evolve drug discovery, diagnostic and therapeutic capabilities[13]. Just one example of clinical need are fragility fractures associated with osteopenia and osteoporosis that account for more hospital bed-days than myocardial infarction, breast cancer, or prostate cancer[14]. These fractures cause high mortality and long-term disability with healthcare cost over $25 billion per year by 2025[15]. Thus, technologies that directly and continuously monitor bone quality[16], and enable exploratory research towards advanced therapeutics in a form factor that enables broad dissemination and a convenient study platform will considerably improve patient quality of life and reduce healthcare costs.

Here we introduce a device platform that uses intimate integration with the osseosurface, the surface of the bone, to enable chronic monitoring of the musculoskeletal system in small and large animal models and lays the foundation for clinical diagnostic tools that can be operated using broadly available near-field communication (NFC) standard in deep tissue. The wireless, battery-free, and fully implantable devices, named osseosurface electronics, can be attached to the surface of bones during orthopedic surgeries and form a chronic interface with bone tissues to directly record a multitude of physiological and biophysical signals critical for the assessment of musculoskeletal health and deliver stimulation in real time (Fig. 1a), providing a powerful point-of-care platform to facilitate rehabilitation and to manage musculoskeletal diseases.

## Results

### Device design
The creation of osseosurface electronics requires several technical innovations that differ from epidermal electronics, such as device footprint and mechanical properties suitable for direct lamination onto the bone surface to minimize mechanical mismatch with the surrounding tissues, and electromagnetic design allowing for direct readout through thick tissues with portable devices to enable smart therapeutics. Special attention to mechanical design of interconnects is required to enable chronic stability of the interconnect and minimize mechanical impact on the targets sensing region to avoid introduction of additional strain. Figure 1b shows a device that meets these design criteria (2.5 cm × 1.5 cm, ~170 mg) and enables direct, conformal lamination to the curved osseosurface with minimal impact on the surrounding tissues. The device features

mechanically isolated biointerfaces that are capable of measuring deformation of the bone with microstrain accuracy, high-fidelity thermography with millikelvin resolution, and photonic stimulation capabilities. The system features a hybrid integration of mechanically compliant flexible substrate with high-performance analog and digital functionalities provided by miniaturized off-the-shelf components (Supplementary Figs. 1 and 2) arranged in a configuration that enables conformality and reliable operation when applied to the curvilinear surface of the bone, as depicted by the layered-device makeup shown in Fig. 1c. The inset of Fig. 1c highlights the multifunctional biointerface comprised of a metal-foil strain gauge, a negative temperature coefficient (NTC) thermistor, and a microscale inorganic light emitting diode (μ-ILED), enabling simultaneous recording of local biophysical signals and on-demand delivery of exogenous stimulation. The device geometry is highly adaptable to accommodate implementation scenario and anatomic structure. Figure 1d shows a device variant applied to a sheep humerus, a testbed to gain insight on capabilities for use as a diagnostic tool after recovery from surgery. We also introduce devices designs for exploratory research in small animal models.

Figure 1e describes the electrical working principle of the system that enables wireless, battery-free operation in a form factor suitable for full implantation that can adopt various NFC chipsets (Supplementary Fig. 3). Near-field magnetic resonant coupling (13.56 MHz, specific absorption rate (SAR) < 20 mW kg$^{-1}$)[12] between an external primary loop antenna (primary antenna) and the on-board loop antenna (secondary antenna) enables, for the first time, reliable radio-frequency (RF) power harvesting through thick tissues up to 11.5 cm with hardware that is compatible with NFC protocols widely available in portable devices[17]. Optimized harvesting electronics matched to digital and analog circuits provide power to an NFC system-on-chip (SoC) that enables operational control through a microcontroller (μC) and NFC transponder in a compact package (4 mm × 4 mm), and analog front-ends (AFE) comprised of passive filters and instrumentation amplifiers that read out thermographic and strain sensors. Design of the sensing circuits is facilitated by circuit simulation (details in the "Methods" section), yielding a strain sensing AFE that consumes 3.24 mW with a sensitivity of 0.194 mV με$^{-1}$, and a thermography AFE that consumes <0.06 mW with a sensitivity of 98.9 mV °C$^{-1}$ (Supplementary Figs. 4 and 5). Digital engineering and manufacturing enable rapid development and deployment of osseosurface electronics in form factors suitable for a variety of operational conditions. Figure 1f shows an example of a system variant that enables operation in a freely moving small animal model without tethers or other externalized elements, which allows advanced exploratory studies. The ultrathin mechanics and low displacement volume (<0.2 cm$^3$) of the device enable rapid recovery of the subject from surgical placement and high-fidelity behavioral studies while collecting bioinformation in real time, which is not possible with existing wire-bound technologies.

### System characteristics
Consistent operation of osseosurface electronics relies on robust wireless power transfer through tissues for large animal models and for therapeutic applications. In the case of small animal models for exploratory research, free motion in a variety of test arenas is required. Both scenarios required robust power transfer and efficient wireless power transfer via magnetic resonant coupling that can be boosted by adopting a secondary antenna with high-quality factor, in our case antennas with high inductance and low impedance[18]. Figure 2a displays a large-animal-model device with optimization though iterative imperial electromagnetic designs (Supplementary Fig. 6) of the secondary antenna using ~185 pF to match the

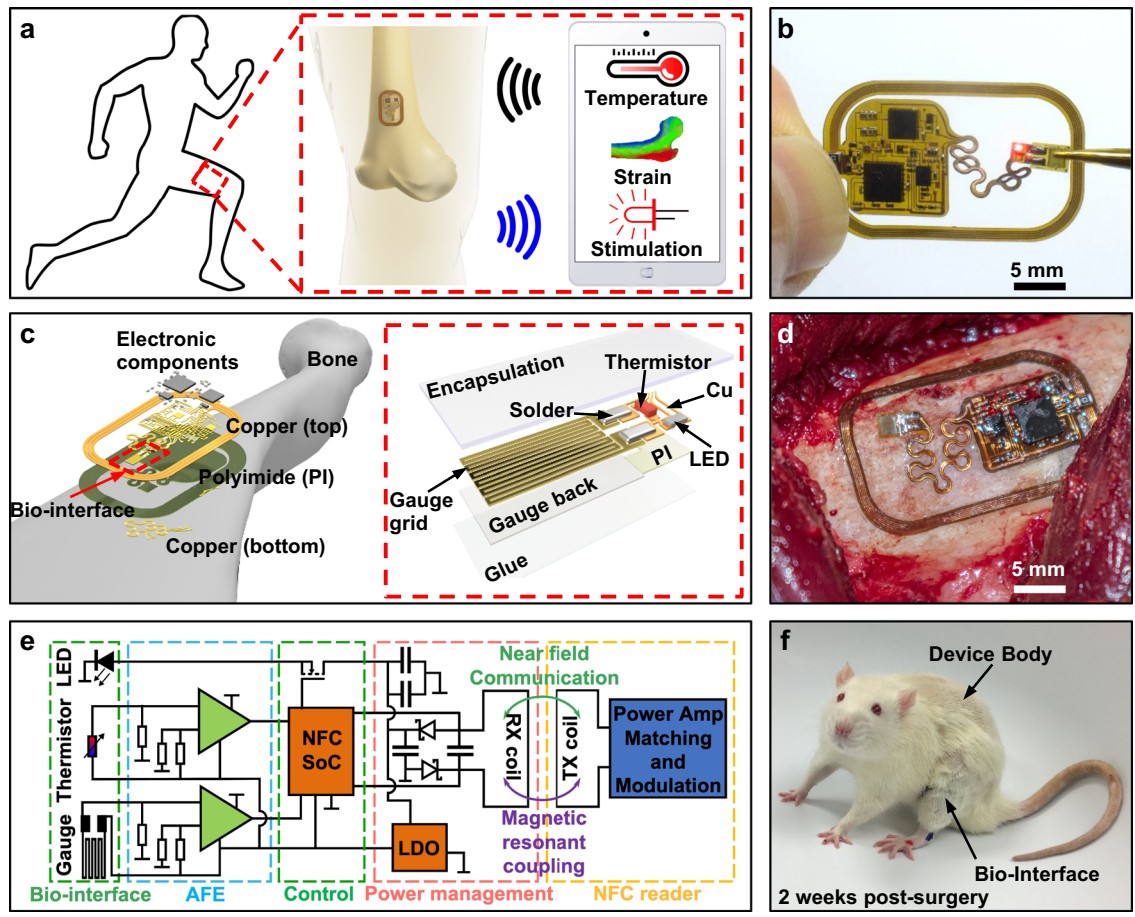

**Fig. 1 Osseosurface electronics: concept, device architecture, and implementation strategies. a** Illustration of osseosurface electronic systems that are permanently bonded to the bone and operate wirelessly to continuously monitor biophysical signals such as bone strain, local temperature, and to deliver optical stimulation to the bone and surrounding tissues. **b** Photograph of an osseosurface device designed for studies in large animal models. **c** Layered makeup of the osseosurface system and its constituent layers. Inset features a close-up view of the multifunctional biointerface comprised of a metal-foil strain gauge, an NTC thermistor, and a μ-ILED. **d** Photograph of an osseosurface device conformally attached on the surface of a sheep humerus. **e** Functional block diagram of osseosurface electronics comprised of an external NFC reader that provides power and facilitates wireless communication, and an implanted system that contains active power management, operational control, analog front-end (AFE), and biointerface. **f** Photograph of a rat (2 weeks after surgery) implanted with an osseosurface device where the main electronics reside on the back while the biointerface is routed and attached on the left femur.

reactance of the antenna inductance of ~745 nH (3 turns, 600 μm wide, 60 μm spacing) at 13.56 MHz allowing for low trace impedance (~30 mΩ)[19], for operational voltages of 2.5 V. As shown by the power-harvesting characteristics (Fig. 2b and Supplementary Fig. 7) measured at the center of a handheld primary antenna (diameter ~20 cm), the maximum values of harvested power (~14 mW) and rectified voltage (~2.1 V) support operation at an electrical load of ~300 Ω. Circular symmetry of the handheld primary antenna (2–8 W of RF power) results in minimal spatial variation in harvested power (Fig. 2c and Supplementary Fig. 8) in close proximity to the antenna and at physiologically relevant distance (5 cm) that corresponds to the depth of implantation on a sheep or human humerus.

To provide power to freely moving small animals, critical for exploratory research in large test arenas (treadmill cage, 45 cm × 12 cm; home cage, 26 cm × 33 cm), devices designed for rat models (Fig. 2d) require an enlarged secondary antenna (3.5 cm × 2.5 cm) and device layout that features serpentine interconnects (~11 cm at full extension) to route the biointerface from the back of the subjects, which houses the electronics and antenna section of the device to the limb that is the sensing target in our experiments. Optimization though iterative imperial

electromagnetic designs (Supplementary Figs. 6 and 7) of the secondary antenna using ~485 pF to match the reactance of the antenna inductance of ~284 nH (2 turns, 600 μm wide, 60 μm spacing) at 13.56 MHz allowing for low trace impedance (~3.3 mΩ)[19], enables harvesting performance of a rectified voltage of ~2.2 V at a load of 300 Ω (16.13 mW) at the center of the 45 cm × 12 cm cage, providing a margin of 0.4 V to enable constant system voltage of 1.8 V throughout all experimental conditions (Fig. 2e). The spatial distribution (Fig. 2f) of the harvested power in the 45 cm × 12 cm cage exhibits sufficient power for stable device operation with primary antenna winding heights at 3 and 6 cm chosen to match device-body implantation location in the freely moving subject (Supplementary Fig. 9). Similar results are obtained for the 26 cm × 33 cm primary antenna, revealing a rectified voltage of ~2.1 V at a load of 300 Ω (14.7 mW) at the cage center and sufficient power at physiologically relevant heights (Supplementary Fig. 10a, b). A comparison between the power-harvesting capability and the power consumption of the biointerface provides a practical guide for feasible in vivo operation modes. Figure 2g shows system power consumption, specifically ~3.24 mW in sensing mode, and ~10.26 mW as optical stimulation is activated, which represents

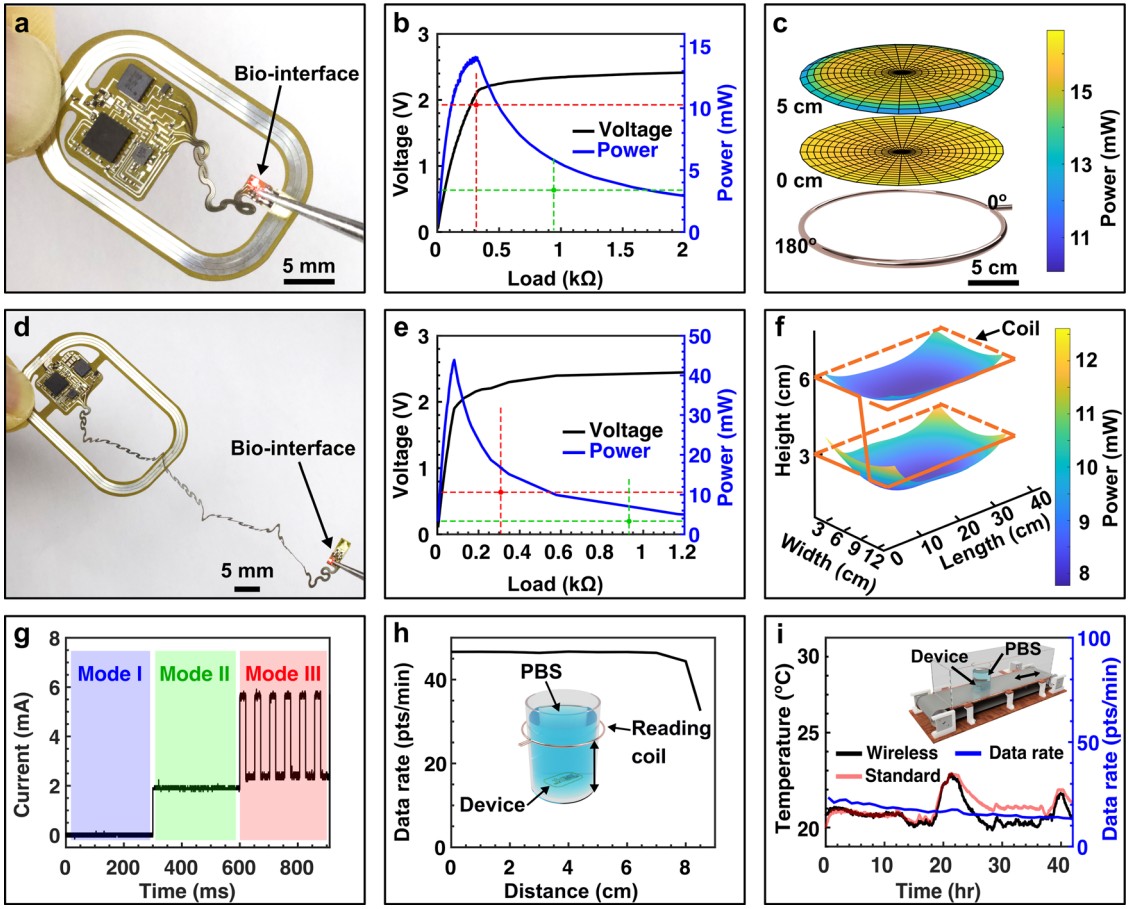

**Fig. 2 System characteristics of osseosurface electronic systems. a–c** Device for large animal models. Device photograph (**a**). Harvested power and voltage as functions of electrical load (**b**). Spatial distribution of harvested power using a handheld primary antenna at a load of 300 Ω (**c**). **d–f** Device for small animal models. Device photograph (**d**). Harvested power and voltage as functions of electrical load (**e**). Spatial distribution of harvested power using a 45 cm × 12 cm primary antenna measured with a load of ~900 Ω (**f**). **g** Power consumption of the device operating at different modes: I. temperature sensing; II. temperature and strain sensing; and III. temperature and strain sensing, and optical stimulation. Modes II and III are represented by green and red dashed lines in (**b**) and (**e**) that indicate the value of electrical load and power consumption. **h** Data rate of wireless communication for a large animal device (immersed in PBS solution) read and powered by a handheld primary antenna as a function of antenna-to-device distance. Inset, 3D rendering of the experimental setup. **i** Demonstration of long-term data recording, for a small animal device (immersed in PBS solution), measured with the 45 cm × 12 cm primary antenna on a custom-built metal-free rat treadmill with back-and-forth motion at a speed of 25 cm s$^{-1}$. The wireless results are benchmarked against environmental temperature recorded by a thermometer (red data line) placed in close proximity. Inset, 3D rendering of the experimental setup.

~60% and ~50% of the harvested power at the corresponding electrical loads, i.e., 900 and 300 Ω, respectively (Fig. 2e). Considering the dependence of power-harvesting capabilities on tilt angle and bending curvature radius (Supplementary Fig. 10c–e), stable and sufficient power can be guaranteed as long as the tilt angle is below 50° and curvature radius is above ~1 cm.

Wireless communication is crucial for implantable devices and often limits device operation in thick tissues[20]. Benchtop experiments reflecting this scenario reveal that the device designed for deep tissue implantation immersed in PBS solution supports a nearly constant data rate of ~46 points per minute with a reading distance up to 7.5 cm from the handheld primary antenna (Fig. 2h) enabling operation as a diagnostic device in large animal models and human subjects. Similar tests designed for use in rodent models reveal uniform data rate in test arenas for rats such as the treadmill cage and home cage (Supplementary Fig. 11). Long-term data recording (Fig. 2i) tests performed in the biologically relevant settings with moving osseosurface electronic devices (details in the "Methods" section) results in stable communication over extended periods of time (42 h, no limitation in operation time) where the recorded temperature

profile matches that of environment temperature measured by a spatially separate thermometer. Further studies of chronic device encapsulation (18 μm Parylene-C) and device operation are conducted via an accelerated rate test in 90, 60, and 40 °C PBS bath to enable device lifetime estimation (205 days at 37 °C) via Arrhenius scaling[21] (Supplementary Fig. 12).

**Biointerface characterization.** Characterization of the multimodal biointerface is performed with benchtop experiments that reflect the in vivo environment. Wireless strain sensors exhibit sensing performance on par with the gold standard wired strain measurement systems in typical physiologically relevant range (0–1200 με)[22], as confirmed by ramping and cyclic loading tests performed on a sheep bone specimen (Fig. 3a–c, details in the "Methods" section), with an estimated sensitivity of ~3.5 ADC με$^{-1}$ and a resolution of ~14.3 με with a sampling rate up to 87 Hz. Cyclic loading tests up to 10 Hz show recording capabilities that surpass gold standard acquisition systems for recording bone strain (Supplementary Fig. 13a–c). Simultaneous recording of temperature and strain with optogenetic stimulation is shown in Supplementary Fig. 13d. Repeatable, hysteresis-free response is obtained upon progressively

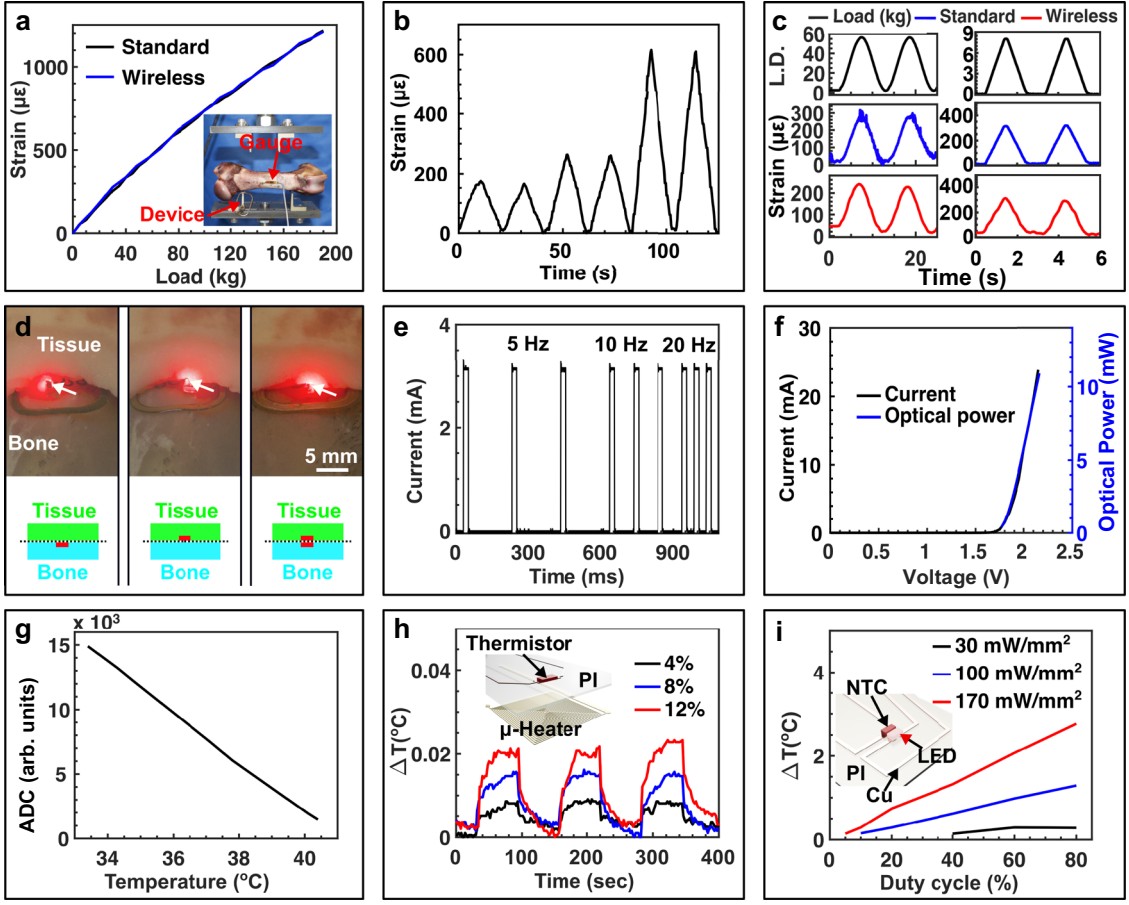

**Fig. 3 Benchtop characterization of the biointerface. a–c** Strain sensing module: comparison of strain–load curves measured using standard wired system and wireless OSE device (**a**); strain profile measured with the wireless device under progressively increasing load (**b**); and strain profiles measured with wired system and wireless device under cyclic load (**c**). Benchtop test setup where the strain gauges are attached on a piece of sheep bone and the load is applied with a MTS system. **d–f** LED module: photographs of devices inserted at the bone–tissue interface, with µ-ILED on the bottom (left), on the top (middle), and on both sides (right) (**d**); current profile of the µ-ILED operating at frequencies of 5, 10, and 20 Hz with pulse width of 20 ms (**e**); current and optical power as functions of driving voltage of the µ-ILED used (**f**). Scale bar in (**d**), 5 mm. **g–i** Thermal sensing module: wirelessly logged ADC values as a function of temperature (**g**); wirelessly measured change in temperature using the NTC thermistor with the micro-heater driven at various levels of power, demonstrating a sensing resolution of <10 mK (**h**); thermal impact of an µ-ILED operating at various optical powers and duty cycles in PBS solution measured wirelessly with a co-located NTC thermistor (**i**). Insets of (**h**) and (**i**), schematics showing the layout of thermistor-micro-heater (**h**) and thermistor-LED front-end (**i**).

increasing loads (Fig. 3b), indicating that a stable bond is formed between the strain gauge and the bone specimen. Reliability of the wireless strain sensor is further demonstrated by cyclic loading with sinusoidal (Fig. 3c), square, and triangle (Supplementary Fig. 14) wave forms.

Current implantable biointerfaces for the musculoskeletal system are typically limited to wired strain sensing[23–27], however our device architecture offers the multimodal integration of stimulation capabilities currently only available in tethered embodiments[28–32]. To demonstrate the feasibility of optogenetic stimulation capability in the periphery[33] on an osseosurface electronic platform, we use miniaturized individually addressable µ-ILEDs to deliver stimulation to the bone and the surrounding tissue. The bone–tissue interface provides a unique platform capable of versatile optical coupling modes tailored for various application scenarios, such as phototherapeutic stimulation for bone regeneration[28] and optogenetic activation of muscular contraction[32]. Figure 3d demonstrates this design flexibility with devices that are capable of illuminating the soft tissue side (left), bone side (middle), and multimodal stimulation (right). Typical operation for optical stimulation covers a parameter space of 5–20 Hz for applications such as sustained tetanic

contraction[31,32], which is achieved on our platform by utilizing a µC for precisely controlled timing as demonstrated in Fig. 3e. The µ-ILEDs used to create an ultraminiaturized form factor (240 µm × 240 µm × 100 µm) and high energy efficiency (>20%, Fig. 3f), ensure minimal thermal impact to the tissues[34].

Continuous monitoring of the local temperature can simultaneously be achieved on our platform with a miniaturized (0201, 0.6 mm × 0.3 mm) NTC thermistor that is integrated monolithically on the biointerface. The wireless temperature sensor exhibits linear response in physiologically relevant temperature ranges (33–41 °C) with a sensitivity of ~1920 ADC K$^{-1}$ and a resolution of 10 mK (Fig. 3g, h, details in the "Methods" section), and can be used to monitor physiological events or to directly assess thermal impact of the µ-ILED, enabling closed-loop control for phototherapeutic and stimulation applications requiring deep tissue penetration. Figure 3i presents the thermal impact of the µ-ILED operating at increasing intensities (30–170 mW mm$^{-2}$) and duty cycles (5–80%) in a physiologically relevant environment mapping the parameter space for optogenetic stimulation and phototherapy. Experimental measurements are in agreement with finite element analysis (FEA) simulation (Supplementary Fig. 15)[34] enabling computational design of the interface. Irradiation that can be

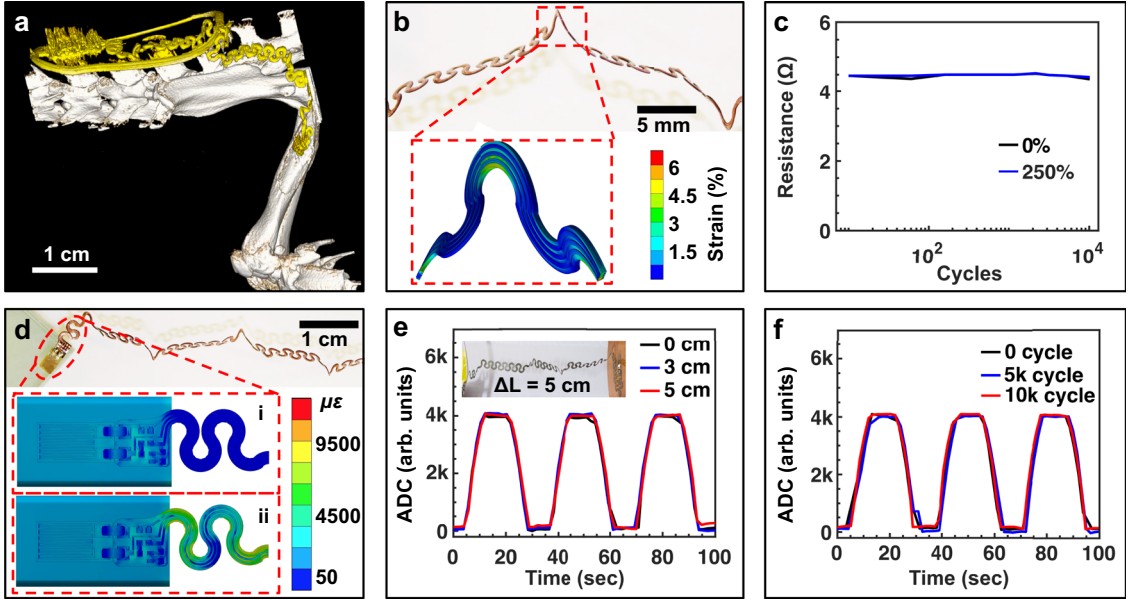

**Fig. 4 Mechanical design and characterization of osseosurface electronic devices designed for rodents. a** Micro-CT scan of a rat implanted with an osseosurface electronic device. The device is highlighted in yellow. **b** Photograph of the self-similar serpentine interconnects stretched to 250% (upper) and FEA strain profile of the serpentine interconnects stretched to 250% (lower). **c** Resistance of the interconnects at 0% and 250% strain during cyclic stretching to 250% for 10,000 cycles. **d** FEA of the biointerface when the bone is compressed by 1000 με while the serpentine interconnects are in two states: i. relaxed (0% strain); and ii. stretched (250% strain). **e** Wirelessly recorded ADC values when the strain gauge is cyclically loaded and unloaded while the serpentine interconnect is strained to 250%, the inset shows the test setup with 250% strain applied. **f** Wirelessly recorded ADC values when the strain gauge is cyclically loaded and unloaded after the serpentine interconnect has been strained for 0, 5000, and 10,000 cycles.

delivered by our platform (e.g., 30 mW mm$^{-2}$) is well beyond the intensity used for optogenetic activation of skeletal muscles[31,32] and evokes only a 0.29 °C temperature increase at a duty cycle of 80%. This suggests operation over a wide range of conditions without thermal activation of neuronal function.

**Mechanical characterization.** Subdermal implantation in small animal models involves placement in highly mobile areas that require a compact formfactor with mechanical design strategies that can withstand repeated strain cycles without interfering with biosensor readouts. This is evident when studying the micro-CT scan (Fig. 4a) of a rat with an implanted osseosurface electronic device, where the device body is anchored subdermally around the lumbar vertebra and connection to the biointerface, which is located on the left femur, is accomplished via self-similar serpentine interconnects[35]. The durability of the serpentine interconnect is critically important for reliable in vivo operation. FEA simulation-guided design enables reliable performance under repeated strains of over 250% while maintaining a maximum strain of ~0.7% in the copper traces which is below 1%, the failure strain of copper (Fig. 4b)[35]. Chronic electro-mechanical stability is confirmed by cyclic straining to 250% for 10,000 cycles, with negligible change in conductivity (Fig. 4c). Critical to the stability of sensor readings during behavior is the mechanical isolation of the stretchable interconnect and the strain sensor, this includes introduction of strain to the bone during deformation of the serpentine interconnect. The biointerface is mechanically isolated by the serpentine interconnects designed to transfer minimal strain during deformation to the bone and the strain sensor biointerface, as confirmed by FEA in Fig. 4d (Supplementary Fig. 16) showing that stretching the serpentine by 250% does not influence the sensitivity or accuracy of the strain gauge (Fig. 4e, f and Supplementary Fig. 17, details in the "Methods" section). Further evaluation of serpentine electro-mechanics are performed with benchtop experiments using a servo-hydraulic materials

testing system (MTS) indicating stable operations of recording capabilities under multiple cycles of high strains (Supplementary Fig. 18) above physiological strains as means to evaluate robustness of the approach. Both benchtop electro-mechanical testings of the monolithic serpentine structure with FEA simulations of the serpentine show solid performance in strain isolation enabling flexible sensor placement to distal regions with minimal effect of sensor readouts.

**In vivo studies in rodents.** The subdermally implantable form factor and wireless, battery-free operation enable an in vivo multimodal bi-directional interface with the musculoskeletal system without compromising the free motion of subjects in various test arenas (Fig. 5). The implantation of osseosurface electronics in rats involves a skin incision on the back of the subject, device placement into the subcutaneous space, subcutaneously tunneling the biointerface to the limb, attaching the biointerface to the femur with cyanoacrylate, and closing the skin with resorbable sutures (details provided in the "Methods" section and Supplementary Fig. 19), following the University of Arizona Institutional Animal Care and Use Committee approved protocol (19–572 IACUC). Devices are tracked and monitored after surgery to gain insight into failure mechanisms. Majority of device failures examined after explanation indicate defects in the Parylene-C encapsulation induced by surgical tools (Supplementary Table 1). Mitigation of these issues in later experiments utilized sutures to manipulate and position the biosensor, resulting in lower failure rates. The optoelectronic interface attached on the bone surface allows for direct optical stimulation of skeletal muscles in freely moving subjects, as shown in Fig. 5a where illumination of the muscle is visually validated, highlighting a systematic advantage over traditional transdermal light sources[30] or optical fiber-based techniques[29], which are challenging to implement in highly mobile areas. Light delivery can be programmed to control frequency and duty cycle of stimulation

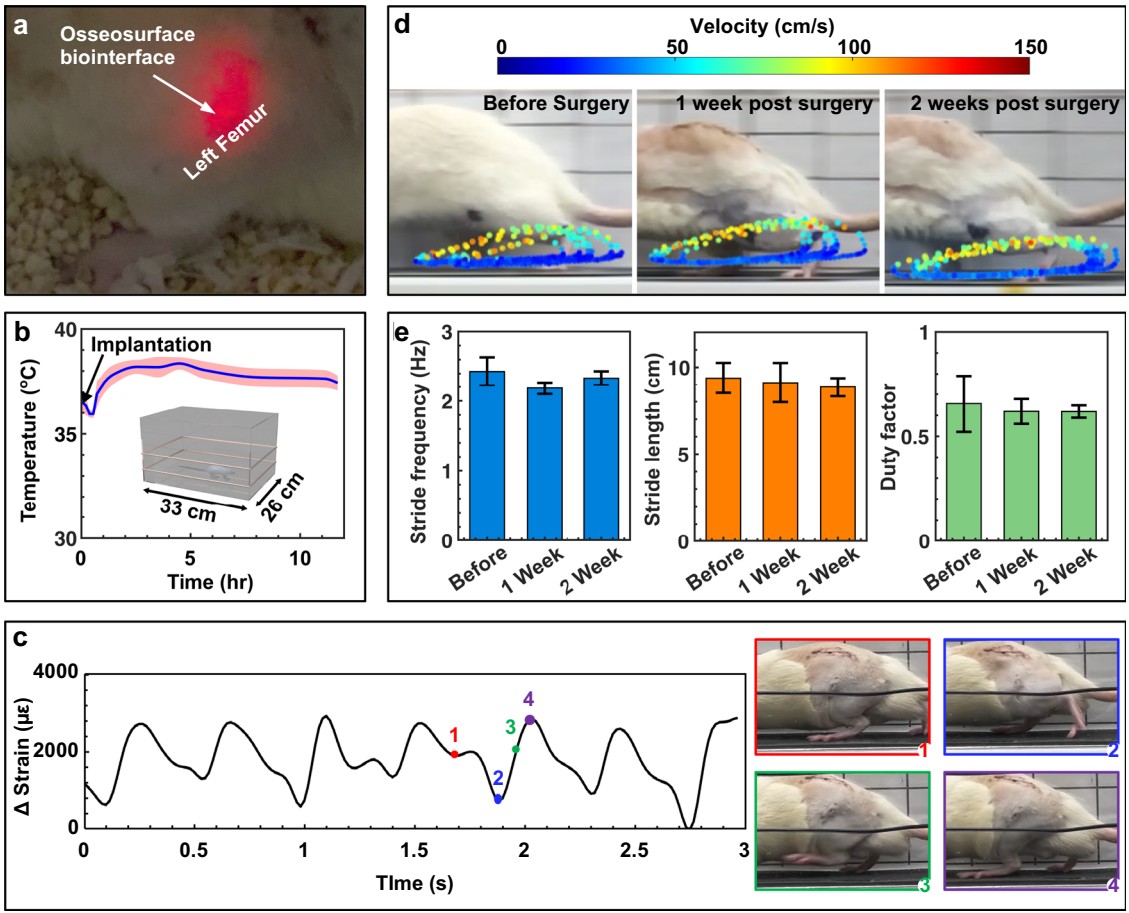

**Fig. 5 In vivo studies of OSE in rats. a** Photograph of the rear part of a rat featuring the µ-ILED attached on the femur illuminating through muscle and skin. **b** Temperature profile recorded in vivo during the period of ~11.5 h following the implantation surgery. Inset, schematic of the rat residing in the home cage. **c** Time-synchronized strain recording and corresponding video frames of the rat gait cycle. **d, e** Comparison of the rat's gaits at different stages of the study: before surgery, 1 week after surgery, and 2 weeks after surgery: photograph of rats walking on the treadmill overlaid with the trajectory of the ankle. Color of the dots represents the velocity of the ankle (**d**). Key parameters: stride frequency (before: mean = 2.4293, SD = 0.4057, $n = 7$; 1 week: mean = 2.1843, SD = 0.1692, $n = 7$; 2 week: mean = 2.3336, SD = 0.197, $n = 7$); stride length (before: mean = 9.3812, SD = 1.749, $n = 7$; 1 week: mean = 9.1175, SD = 2.2686, $n = 7$; 2 week: mean = 8.8466, SD = 1.0577, $n = 7$); and duty factor (before: mean = 0.6558, SD = 0.2689, $n = 7$; 1 week: mean = 0.6189, SD = 0.118, $n = 7$; 2 week: mean = 0.6182, SD = 0.0585, $n = 7$) that characterizes the rat gait before and after implantation (**e**).

each with 16-bit precision. Chronic functionality is demonstrated with in frame analysis of tracked red pixel intensity over time (Supplementary Fig. 20) from video recordings 22 days post implantation (Supplementary Video 1). In vivo device operation of temperature-sensing capabilities is tested over extended time periods demonstrated by a 10 min moving window of thermographic recording that show recovery of the animal post-surgery for over 11 h in the home cage (blue) with a standard deviation within the 10 min moving window (pink) (Fig. 5b). Following the surgical implantation of the device, a dip in local body temperature during the first half-hour of recovery after implantation is observed, which is an expected hypothermic response to isoflurane anesthesia[36] and transfer from a warmed surgical table to a cooler recovery area. Following recovery from anesthesia, the limb temperature remained within the published range of normal body temperature for male Sprague Dawley rats[36]. Strain profile recorded wirelessly in a custom treadmill arena with the 45 cm × 12 cm primary antenna reveals characteristic loading and unloading phases with absolute strain values comparable to literature values for the rat femur obtained by tethered sensors[22]. Time-synced wireless recording of strain are compared with video recording of the animal during walking periods after a week of recovery (Supplementary Video 2). Analysis enables intimate

insight into bone strain of the femur during gate. The recordings of strain show four distinct points during a gate cycle: (1) mid-stance, (2) lift-off, (3) mid-swing, and (4) touch-down[37,38]. These points and their corresponding frames are shown in Fig. 5c. Behavioral assays, gait analysis in particular, are useful tools to investigate pathogenesis and develop new therapeutics for musculoskeletal disorders such as osteoarthritis[39]. Therefore, a basic requirement is that investigative devices do not alter gait performance of the subjects, which is not easily achieved with tethered approaches. Due to the low-profile and mechanical compliance, osseosurface electronic implants do not affect the subject's gait, as revealed by deep neural network analysis of rats with implants (Fig. 5d, e, details in the "Methods" section). The heat maps displayed in Fig. 5d indicate that the trajectories of the paw of naive and experimental subjects with implants remain qualitatively the same and the spatial distribution of paw velocity is unchanged at various stages of the study. Spatiotemporal gait characteristics, including stride frequency, stride length, and duty factor[40], presented in Fig. 5e, provide quantitative evidence that normal gait is sustained after implanting the osseosurface electronic device. The welfare of the test animals is also reflected by the steady weight-gain from 2 days post-surgery until the study is terminated (Supplementary Fig. 21). In addition, histological

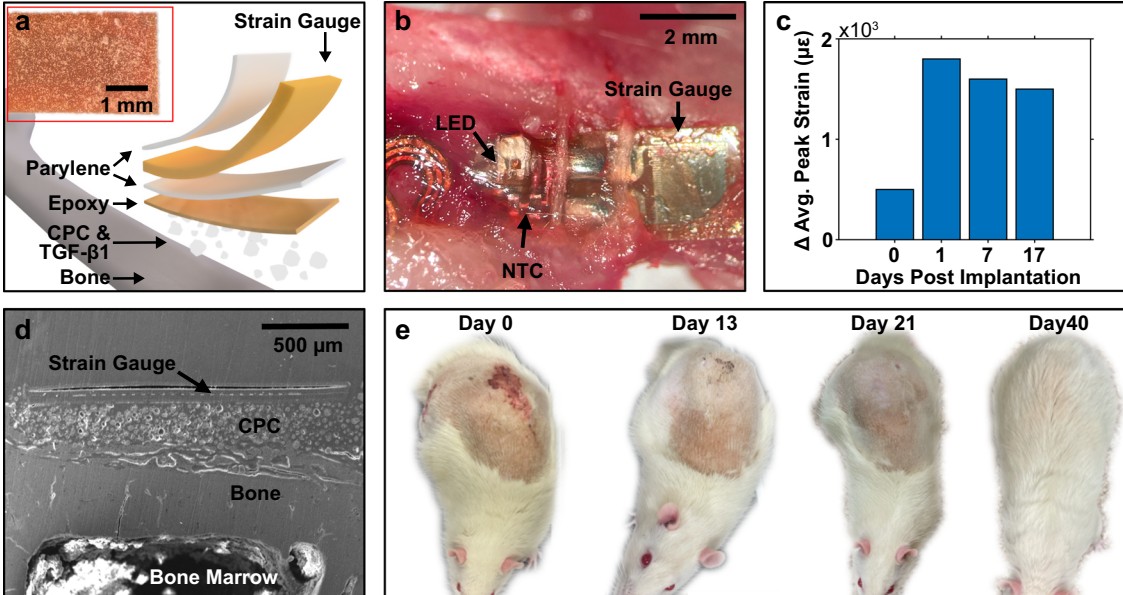

**Fig. 6 Promotion of osteogenesis using surface-engineered calcium phosphate ceramic coatings. a** Photograph of the bottom coated with CPC and a side view diagram of layer composition of the strain gauge sensor with CPC. **b** Image of the device 3 weeks post implantation. **c** Wireless strain gauge data collected over 2 weeks post implantation showing stable CPC adhesion. **d** Single electron microscope image of cross-section of CPC adhesion to bone. **e** Animal healing after implantation.

analysis shows that the subcutaneous implantation of the device body in the lumbar vertebra region, is surrounded with fibrous tissue containing blood vessels without any evidence of inflammation or significant foreign body response after 2 weeks of in vivo operation (Supplementary Fig. 22).

**Permanently attached osseosurface electronics.** The ultrathin electronic platform enables direct lamination onto the bone, which provides the opportunity to attach devices permanently to gather information on bone health long term. Because of cell turnover, any glue will exhibit limited lifetime and continuously degrading biointerface quality. A solution to this problem is to directly grow osseosurface devices to the bone using calcium phosphate ceramic (CPC) particles[41]. For the use in animal subjects, fast adhesion is critical to enable accelerated experimental timelines, which can be accomplished by the addition of transforming growth factor beta 1 (TGF-β1), an osteogenic protein that enables rapid bone bonding to the CPC particles[42]. We use this scheme to bond CPC particles to the osseosurface electronics using implant-grade epoxy, and subsequently applying TGF-β1 to the CPC particles. The resulting layered-device composition is shown in Fig. 6a. The exposed particles enable effective bone bonding after implantation and temporary fixation with resorbable sutures (Fig. 6b). The efficacy of this approach can be observed in vivo by recording the average delta strain values of gait on a treadmill. Figure 6c shows the evolution of the delta strain over the course of several weeks, here we can observe weak bonding of the stain gauge to the bone for the day of the surgery with rapid increase in delta strain values in the following days and a stable attachment after a week (Supplementary Fig. 23). The successful growth of the device to the bone can also be observed after explanation (day 27) and cross-section preparation, and subsequent secondary electron imaging (Fig. 6d and Supplementary Fig. 24; details in the "Methods" section). The micrograph shows successful growth of the bone to the particles that permanently affix the flexible electronics enabling chronic recording of bone health. The chronic application of these devices enables a multitude of new studies, for example, strain-mediated

bone remodeling with experimental paradigms that involve extensive mobility of subjects and longitudinal studies with multiple cohabitating subjects, which is facilitated by complete recovery after implantation of the wireless and battery-free device as shown in Fig. 6e.

**In situ studies in large animal models.** The scalability of our platform enables the use in large animal models with minimal modifications. Immediately after euthanization, following the University of Arizona Institutional Animal Care and Use Committee approved protocol (16–202 IACUC), in situ operation of devices designed for large animals are demonstrated on sheep humeri where small footprint and soft mechanics of the device enable conformal application to the bone surface (Fig. 7a and Supplementary Fig. 25a, b). The thermography function allows for continuous monitoring of local temperature as an important indicator of subject health throughout the surgical procedure. As shown in Fig. 7b, distinct features can be identified and correlated to events such as closing and reopening of the incision. Devices implanted deep (>5 cm) in the body are warmed significantly faster to a saturation level close to the core body temperature (Supplementary Fig. 25c, d) than those implanted shallower (~2 cm). The wireless strain recording capability through thick tissues is validated while the humerus is loaded in three-point bending (Fig. 7c), successfully capturing the bending events with well-discernable loading and unloading phases despite considerably lower strain than those noted during gait measured from active sheep[43].

## Discussion
Wide dissemination of osseosurface electronics requires an implementation strategy that can be easily integrated into existing orthopedic surgical procedures. We demonstrate a device attachment strategy that is assisted by a 3D-printed applicator and can be accomplished within 10 min, minimally altering existing surgical protocols[44,45] (Fig. 7d, details in the "Methods" section). The applicator can be customized to match localized anatomic structure and device dimensions (Supplementary

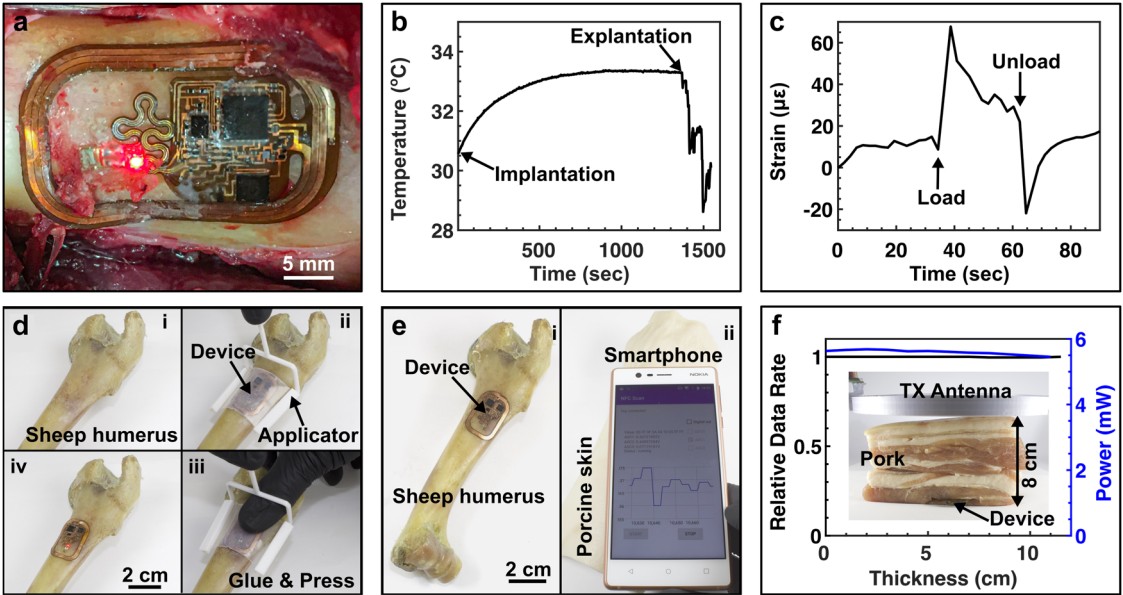

**Fig. 7 In situ studies in sheep and diagnostic application. a** Photograph of a wirelessly powered osseosurface electronic device attached on a sheep humerus. **b** Temperature profile during implantation surgery on sheep humerus recorded wirelessly through tissue. **c** Strain profile of 3-point bending test of sheep humerus recorded wirelessly through tissue. **d** Tools and strategy to attach osseosurface devices on the bone: i. bone surface is cleaned by sanding; ii. device with applied glue is brought into contact with the bone surface with applicator; iii. device is pressed firmly with finger; and iv. applicator is removed. **e** Photos of demonstration of real-time signal recording through porcine skin (~1 cm thick) using an NFC-enabled smartphone: i. device attached on the sheep humerus; and ii. device covered by porcine skin and real-time signal readout with NFC-enabled smartphone. **f** Relative data rate and power availability as functions of tissue thickness. Inset shows photograph of experimental setup.

Fig. 16), minimizing the impacts on the surrounding tissues, and providing a path towards implementation as a diagnostic tool following routine surgical procedures.

Easy readout with industry standard RF protocols and rapid attachment with chronic interfaces enable osseosurface electronics to provide significant opportunities for the direct measurement of crucial indicators of bone health in real time. Consequently, this allows a point-of-care solution to monitoring post-fracture rehabilitation and managing musculoskeletal conditions. Here, we demonstrate the successful device operation using an NFC-enabled smartphone through tissues, as shown in Fig. 7e, where the sensor signals from a device covered by a piece of porcine skin (~1 cm thick) simulated human tissue on the tibia are directly visualized in real time as the smartphone is used to provide power to and wirelessly communicate with the device (Supplementary Video 3). These results demonstrate the feasibility of operating osseosurface electronics with a smartphone in an at-home setting. Locations covered with several muscle layers such as the femur in larger animals, can be accessed with a dedicated reader solution with near-field-enabled clothing for efficient power transfer and continuous communication during everyday behaviors[46]. Figure 7f plots the relative data rate and harvested power as functions of tissue thickness between the reader and the device, showing no degradation in either data rate or available power level at the devices at tissue thickness up to 11.5 cm.

This highlights successful wireless power transmission and data communication with a battery-free and soft implantable device through thick tissues[47,48], which enables implantation on almost any location on the human skeletal system.

The miniaturized form factors, soft mechanics, versatile sensing/stimulation options, as well as robust wireless power-harvesting and communication capabilities make osseosurface electronics a powerful platform to establish direct and chronic bidirectional interface with the musculoskeletal system. This offers unprecedented opportunities for mechanistic studies of osteogenesis and pathogenesis of musculoskeletal diseases, as well as the development of new types of diagnostics and therapeutics. Furthermore, we demonstrate the successful attachment of the osseosurface electronic biointerface to the bone utilizing surface-engineered CPC particles, an important aspect of this technology that enables recording on the order of many years enabling device operation for the lifetime of the subjects without requiring secondary surgery. This chronic platform can then enable acquisition of holistic data of bone health status and closed-loop therapeutic intervention to facilitate treatment and rehabilitation.

## Methods

**Device fabrication.** Flexible circuitries were fabricated by UV (355 nm) laser ablation (ProtoLaser U4, LKPF, Germany) using a sheet of copper-clad polyimide foil (Dupont, Pyralux AP8535R, copper /Polyimide/copper, 17.5 μm/75 μm/ 17.5 μm) as substrate. Subsequent sonication in solder flux and isopropanol removed the surface oxides formed during laser ablation. Surface-mount components, including passive components such as resistors (0201, 0.6 mm × 0.3 mm), capacitors, Schottky diodes (Skyworks Inc.), Zener diodes (Comchip Technology Corp., 5.6 V), μ-ILED (red, ES-AEHRAX10, EPISTAR), and NTC thermistor (NTCG064EF104FTBX, TDK), as well as active components such as MOSFET (PMZ130UNE, Nexperia), low-dropout linear regulators (LDO, TCR2DG18, Toshiba), instrumentation amplifier (INA, AD8235, Analog Devices), and NFC SoC (RF430FRL152H, Texas Instruments), were manually placed on the flexible circuit and reflowed with low-temperature solder (Indium Corp.). A metal-foil strain gauge (N2A-06-S5182N-10C/E4, Micro-Measurements) was then integrated by using a chisel-tipped soldering iron. Finally, the device was baked at 120 °C in an oven for 30 min to remove residual flux and solvents used in the assembling process. Devices were treated with silane (A174, Sigma Aldrich) and encapsulated with two layers of 9 μm Parylene-C using a Parylene coating system (Parylene P6, Diener electronic GmbH). Devices were then coated with PDMS (Sylgard 184, Dow Corning) by dip coating.

**Circuit simulation.** LTspice XVII was utilized to simulate the electrical characteristics of the strain sensing and temperature-sensing AFE. The circuit diagram and choice of components are shown in Fig. 5. The output voltage of the Wheatstone bridge and instrumentation amplifier were selected as the output parameters. To simulate the basic characteristics of the strain sensor circuit (Supplementary Fig. 5a, b), resistance of the strain gauge was varied sinusoidally from 999 to 1001 Ω with a frequency of 20 Hz. To simulate the strain sensing

performance with various values of the bridge resistors ($R_b$) (Supplementary Fig. 5c, d), $R_b$ was varied linearly from 1 to 10 kΩ, at four values of the gauge resistance (998, 999, 1001, and 1002 Ω). To simulate the basic characteristics of the temperature-sensing circuit, the thermistor resistance was varied from 72.32 to 46.49 kΩ corresponding to the temperature range of 32–42 °C.

### Device characterization

*Wireless power-harvesting capability*. RF power in the range of 2–8 W was provided by a long-range RFID reader module (Feig Electronic GmbH, ID ISC.LRM2500-A). The primary antenna was connected to a tuning/matching circuit board and tuned at 13.56 MHz with a voltage standing wave ratio (VSWR) below 1.5. A 1-turn circular primary antenna with diameter of 20 cm was used to power the devices for large animals, while the devices for rodents were powered by two types of primary antenna: a 45 cm × 12 cm, 2-turn (at heights of 3 and 6 cm) coil that encloses the rat treadmill cage, and a 26 cm × 33 cm, 2-turn (at heights of 3 and 9 cm) coil that encloses the rat home cage.

The completed wireless power-harvesting module was tuned to 13.56 MHz using a spectrum analyzer with reflection bridge (SSA 3032X, Siglent). Capacitors were added to tune the antenna to reach peak attenuation at 13.56 MHz. The capacitors are used to calculate inductance of the antenna by matching the reactance of the capacitor and inductor at 13.56 MHz. Power and voltage characterization of the antenna was tested by placing the device in the center of the test arena in parallel with the arena floor. The rectified voltage was then recorded with a digital multimeter while the electrical load was varied from ~50 Ω to ~5 kΩ.

The spatial distribution of harvested power was measured at a fixed load (300 or 900 Ω), while the device was placed at various locations and different heights (3 and 6 cm) in the test arena. The angular dependence of the power-harvesting capability was measured by varying the angle of the device with respect to the arena floor from 0° to 70° using a rotational jig. The dependence of harvested power on curvature radius was measured by conforming the device to curved surfaces of 3D-printed objects with varying radii of curvature (1.6–3.2 cm).

*Wireless data communication*. Wireless data reading was accomplished by ISOStart (V10.09.00, Feig Electronic GmbH). In order to mimic the in vivo environment, the device was placed at the bottom of a 250 ml beaker filled with 1x PBS (Sigma Aldrich) solution. For the large animal device, data rate was measured while the handheld primary antenna was held at various heights from the device. For the rodent device, data rate was measured at representative locations in the test arenas (Supplementary Fig. 7). Long-term data recording was performed using the rodent device and the 45 cm × 12 cm primary antenna over a ~42 h period while the device was immersed in PBS solution and constantly moving back-and-forth (25 cm s$^{-1}$) on a custom-built rat treadmill. A custom-built thermometer (LMT70, Texas Instruments) was used to monitor the environment temperature. Device lifetime estimation was calculated using an activation energy of 57,800 J mol$^{-1}$ based on device failure at 90 °C and latest measured operation of device at 60 °C. Using the Arrhenius scaling equation, we estimated a device lifetime of 205 days at 37 °C[49].

*Characterization of the wireless strain sensor*. Benchtop tests of the wireless strain sensor were performed using an explanted sheep's femur. The periosteum of the mid-diaphysis was removed from the femur, and the metal-foil strain gauge (N2A-06-S5182N-10C/E4, Micro-Measurements) of the wireless osseosurface system was attached to the femur using a cyanoacrylate-based adhesive (M-Bond 200, Micro-Measurements). A wired strain gauge was subsequently bonded on top of the wireless gauge following the same procedure using a stereomicroscope to ensure overlap and alignment of the sensing elements. The sheep femur was loaded in four-point bending configuration using a servo-hydraulic MTS (Series 810, MTS Systems Corporation) while recording load and strain from the wired sensors using standard data-acquisition system (System 8000 and StrainSmart, Micro-Measurements). Measurements from the wireless gauge were recorded with a handheld antenna using the Feig reader and ISOStart. Various load profiles were tested, including a linear ramp load, and cyclic loading with a sinusoidal wave pattern, a square wave pattern, and a triangle wave. The femur was loaded to a peak load of 190 kg at rates ranging from 5–60 kg s$^{-1}$ in each profile.

*Characterization of the wireless optical stimulation module*. The current–voltage ($I$–$V$) characteristics of the μ-ILED was recorded with a source measurement unit (SMU, Keithley 2450) operating in the linear sweeping mode. The optical power was measured with an integration sphere (OceanOptics FOIS-1). The current consumption of the μ-ILED was measured with an Oscilloscope (Siglent SDS 1202X-E) measuring the voltage across a 10 Ω resistor in series with the μ-ILED. The wireless circuit uses a MOSFET (PMZ130UNE, Nexperia) to drive the μ-ILED with programmed predefined frequencies and duty cycles stored in the μC (ATTiny 13 A, Microchip Technology). Total current consumption of the device was measured using a benchtop power supply (1.8 V) with an Oscilloscope (Siglent SDS 1202X-E) recording the voltage drop across a 10 Ω resistor.

*Characterization of the wireless thermography*. The wireless temperature sensor was immersed in a water bath whose temperature was varied from 33 to 41 °C using a hotplate and monitored by a commercial thermocouple digital thermometer. The

sensor signal was wirelessly recorded with a handheld antenna using the Feig reader and ISOStart and used for sensor calibration.

In order to characterize the resolution of the wireless thermographic biointerface, the copper on the bottom side of the Pyralux substrate was laser ablated to form a micro-heater beneath the NTC thermistor (design of the micro-heater is shown in Supplementary Fig. 9a). The micro-heater was wirelessly powered by the osseosurface device and the on-board voltage regulator was used to drive PWM controlled 1.8 V with a heater element resistance of 5.36 Ω that was characterized prior with a SMU (SMU, Keithley 2450) enabling a defined heater output. The μC was used to control a programmed sequence of duty cycles varying from 4% to 12% to control the MOSFET that drives the micro-heater. The fully encapsulated circuit was immersed in a water bath, and a handheld antenna was used to wirelessly power the device and retrieve temperature recordings from the NTC. The resolution of the temperature sensor was subsequently determined by the smallest sensor response that could be distinguished from the background noise.

A circuit with co-located NTC thermistor and μ-ILED (~0.1 mm apart) was fabricated and used to demonstrate the capability to directly measure the thermal impacts of optical stimulation. The thermistor and μ-ILED were immersed in a PBS bath to mimic the in vivo environments and to prevent the device from overheating at high optical power. A function generator (Siglent, SDG 1032X) was used to drive power to the μ-ILED to test μ-ILED heating of surrounding tissue using a micro-temperature sensor collocated by the μ-LED varying voltage (1.85, 2, and 2.12 V), frequency (5–40 Hz), and duty cycle (5–80%), while the temperature sensor signal was wirelessly read out with a handheld antenna.

*Mechanical durability of serpentine interconnects*. The serpentine interconnects were mounted on a custom-built stretching stage and subjected to ~250% strain and stretched cyclically for 10,000 cycles. The resistance of the serpentine copper traces was measured with a digital multimeter.

The strain gauge of an osseosurface electronic system was bonded onto a piece of Kapton foil (~75 μm thick). The Kapton foil was mounted on the stretching stage and subjected to cyclic bending with a radius of curvature of ~2 cm, while the serpentine interconnects were stretched to various lengths (ΔL ~3 and 5 cm) with a separate stage. The strain sensor signal was wirelessly recorded with a circular primary antenna (20 cm in diameter). The same measurement was repeated after the serpentine had been stretched for 5000 and 10,000 cycles.

### Mechanical simulation

Ansys® 2019 R2 Static Structural was utilized for static-structural FEA simulations to study the strains induced in the copper traces of the serpentine interconnects, and the effectiveness of mechanically isolating the strain gauge from other parts of the device. The components of the devices, including the copper and constantan traces, Pb-free solder, polyimide (PI), and Parylene-C encapsulation layers, were modeled using the layouts used in device design. The mechanical properties (Young's Modulus ($E$) and Poisson's Ratio ($v$)) used for the simulation were: $E_{PI} = 4$ GPa, $V_{PI} = 2.7579$, $E_{Cu} = 121$ GPa, $V_{Cu} = 0.34$, $E_{Constantan} = 162$ GPa, $V_{Constantan} = 0.32$, $E_{Parylene} = 2.7579$ GPa, $V_{Parylene} = 0.4$, $E_{Solder} = 43$ GPa, $V_{Solder} = 0.29$.

*Simulation of strain in serpentine interconnects*. The model was simulated using the following meshing parameters: program-controlled nonlinear mechanical elements with an element size of $8.0 \times 10^{-5}$ m, a body sizing insert condition with element size $1.0 \times 10^{-5}$ m applied to all the traces, and a body sizing insert condition with element size $2.0 \times 10^{-5}$ m applied to the PI layer. The simulation was performed by fixing one end of the selected serpentine segment and applying a displacement of 5.75 mm upwards to the other end as shown in Supplementary Fig. 15, producing a deformation replicating that observed in benchtop testing of the serpentine interconnects (Fig. 4b).

*Mechanical isolation of the strain gauge*. The model was simulated using the following meshing parameters: program-controlled mechanical elements with an element resolution of 2, and a body sizing insert condition with element size $2.0 \times 10^{-5}$ m applied to all the traces. Two simulations were performed, both with 1000 μɛ applied to the bone model by fixing one end while displacing the other end by $7.9 \times 10^{-6}$ m, as shown in Supplementary Fig. 15: (i) no strain was applied to the serpentine interconnects; (ii) the serpentine interconnects were subjected to 3D displacements ($x$: 1 mm, $y$: 3 mm, and $z$: 3 mm; details in Supplementary Fig. 10b).

### Animal studies

All animal experiments were performed following a University of Arizona Institutional Animal Care and Use Committee (IACUC) approved protocol (Sheep: 16–202, Rat: 19–572). For the in vivo study in rats, five male 450 g Sprague Dawley rats were used. The implanted devices were sterilized using ethylene oxide and aerated for 24 h prior to placement. Rats were anesthetized using isoflurane and were given a subcutaneous injection of 1.0 mg kg$^{-1}$ Buprenorphine SR prior to surgery. A 2 cm incision was made along the midline of the back over the lumbar spine. A 1 cm incision was made over the lateral thigh, and the anterior surface of the femur was exposed subperiosteally through a lateral approach. The device was placed in a subcutaneous location through the back. Passage of the strain gauge into the thigh via a subcutaneous tunnel was facilitated

by the serpentine interconnect. The strain gauge was fixed to the femur using a cyanoacrylate adhesive (M-Bond 200, Micro-Measurements). For implantation of CPC-coated devices in small animals, a 5-0 vicryl suture lassoed around the biosensor was used to pass the sensor from the lumbar region through the subcutaneous tunnel to the thigh. This lasso technique was used to avoid manipulation of the device with sharp surgical instruments and minimize the risk of damage to the encapsulating layers. CPC-coated gauges were secured to bone using two 5-0 vicryl sutures without adhesive. Layered closure was performed using 5-0 vicryl for fascia and 4-0 Quill for subcutaneous tissues prior to recovery from anesthesia.

During recovery, strain and temperature measurements were continuously recorded with the 26 cm × 33 cm primary antenna while rat behavior was recorded via a webcam. After 2 days, measurements were collected while rats walked on a custom-built treadmill at 25 cm s$^{-1}$. Sensor signals were recorded wirelessly with the 45 cm × 12 cm primary antenna while high-speed (1920 × 1080, 120 frame per second, iPhone 6 S) videos were recorded with a camera placed ~50 cm from the treadmill for the purpose of deep neural network of the gait. Before each exercise session, the rat received a minimum of 20 min of habituation to treadmill. Strain changes were observed during loading and unloading of the femur (Fig. 5c). After 2 weeks the rats were euthanized. Following euthanasia, device placement was characterized by scanning rats at 20 μm resolution using a Siemens Inveon micro-CT Scanner. For histology, tissue surrounding the implanted device was excised and imaged using an optical microscope (Wild M3Z Stereozoom Microscope, Leica) with an attached camera (iPhone 12 Pro, Apple) (Supplementary Fig. 22a–c), fixed in 10% formaldehyde for 24 h and embedded in paraffin. Ten-micron sections were cut and stained with hematoxylin and eosin (Supplementary Fig. 22d–f). Slides were viewed using a Nikon microscope coupled to a Nikon DS-Fi1 camera at magnifications ranging from 20x to 100x.

*In situ study in sheep.* One male 2.5-year-old sheep was used to confirm function of the device in a large animal. The sheep was a control used in another study and had a 4.2 cm femoral defect created 6 months prior to use. Device placement occurred following administration of euthanasia medications for the other study and was completed prior to significant changes in body temperature. The anterior surface of the mid-diaphyseal humerus was exposed subperiosteally through a lateral approach. The wireless device was fixed to the anterior surface of the humerus deep to brachialis. The facia and skin were closed with running sutures. Measurements of strain and temperature were collected continuously using a handheld antenna placed on the skin for 20 min following surgical placement of the device. Function of the strain sensor was confirmed by loading the sheep's humerus in three-point bending (Fig. 6c).

**Deep neural network analysis.** DeepLabCut (version 2.2.b6) was used to perform deep neural network analysis. The neural net was trained with a 1 min video clip where 200 frames were extracted as training material. The training session was performed with 200,000 iterations on the high-performance computer of University of Arizona. After training, a 3 s video clip with consistent gait was tracked and analyzed by the software to extract the coordinates of the left hind paw for each frame. The timestamp and coordinates were subsequently utilized to calculate the spatiotemporal gait characteristics, including stride length, stride frequency, duty factor, and the velocity of the paw.

**Osseosurface electronics bone attachment.** An applicator comprised of a 3D-printed frame and an elastomeric membrane (Supplementary Fig. 26) was utilized to facilitate fast attachment of the device on the bone surface. Before attaching the device, the surface of the bone was abraded. The device was first attached to the applicator with the top side in contact with the elastomeric membrane. Adhesive agent (M-Bond 200, Micro-Measurements) was then applied to the bottom side of the device. The applicator was designed to make surgical process in large animals easier by improving the ability to conformally mount the strain gauge to the bone in one step. Once placed, the device was then firmly pressed against the bone with an even force until the adhesive agent cured. Finally, the applicator was carefully peeled off with minimal out-of-plane sheer force between the membrane, device encapsulation, and adhesive agent.

**Calcium phosphate ceramic coating.** Implant-grade epoxy (Master Bond EP42HT, Master Bond, Inc., Hackensack, NJ) was prepared according to the manufacturer's instructions and a thin even layer was blade coated to the bottom surface of the strain gauge (N2A-06-S5182N-10C/E4, Micro-Measurements). Spherical crystalline calcium phosphate ceramic (CPC2, CeraMed) was added to cover the surface of the epoxy and silicone rubber was used to gently apply downwards pressure on the CPC particles while curing the epoxy for 24 h at room temperature. All CPC coatings were inspected for surface exposure without epoxy coating to enable bonding to the bone. Devices were coated with TGF-β1 to accelerate bone-to-CPC bonding.

**Scanning electron microscope imaging of CPC–bone adhesion.** A rat after 27 days post implantation was euthanized, and the femur was explanted along with the attached strain gauge. The bone was cleaned, removing any soft tissue and fixed using a neutral-buffered formalin for 2 h. The bone was dehydrated in a solution of ethanol (EtOH) starting at 70% for 2 h and increasing concentrations for at least

2 h as follows: 80% EtOH, 95% EtOH, 95% EtOH, 100% EtOH, 100% EtOH, xylenes, and 100% EtOH. The bone was embedded in methyl methacrylate (Technovit 9100 kit, Kulzer) through a pre-infiltration process for 2 h at room temperature and an infiltration process within a vacuum chamber for 15 min to remove any bubbles. The embedded bone is then cured at −4 °C over night. The bone and strain gauge were cut exposing the cross-sectional view. The cross-section surface was smoothed and polished using a grinding wheel (GP-25, Leco) from 120 to 600 grit. The surface was gold sputtered (Hummer 6.3, Anatech USA) allowing for a gold thickness of 9 nm. SEM images were taken with various magnifications (60x, 120x, 280x, 300x, and 600x) around the bone–strain gauge interface using (Inspect S50, FEI) operating at 30 kV.

**Device operation in deep tissue.** Slits of ~3 cm length were cut into the porcine tissues and layers of tissue were stacked to increase device–primary antenna distance, as shown in Supplementary Fig. 27 for a test at 8 cm device–primary antenna distance. The rectified voltage and data rate were recorded with a handheld primary antenna while the device was loaded with 1 kΩ and inserted into the slits to result in device–primary antenna distances from 0 to 11.5 cm.

*Real-time data visualization through tissues.* A device was attached on a sheep's humerus following the device attachment procedures described above, and subsequently covered by a piece of porcine skin (~1 cm thick). An NFC-enabled smartphone running a custom software was brought in proximity to the porcine skin. The NFC connection was subsequently established, and the sensor signals were visualized on the smartphone in real time (Supplementary Video 3). To demonstrate real-time readout, a device was attached onto a model of a human femur, then covered with a piece of porcine skin, and subsequently operated through the tissue with a smartphone while the femur model was strained to induce signal change (Supplementary Video 4).

## Data availability
All data needed to support the conclusions presented in this paper are available in the manuscript and/or the Supplementary Information.

## Code availability
Custom MATLAB code to decode wireless recording and data to support this study have been deposited in the Center for Open Science database that can be accessed at [https://osf.io/ucbjh/][50].

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

## Acknowledgements

We acknowledge the support from University of Arizona Department of Biomedical Engineering startup funds, the Improving Health Technology and Research Initiative Fund (TRIF), the Craig M. Berge Faculty Fellowship and Core Facilities Pilot Program (CA-CFPP NANO-3310342) (P.G.). We acknowledge support from the National Institute of Biomedical Imaging and Bioengineering of the National Institutes of Health T32EB000809 and the ARCS Foundation Spetzler Endowment (A.B.). We acknowledge support from the Herbold Fellowship (K.A.K.).

## Author contributions

L.C., A.B., J.A.S., D.S.M. and P.G. designed research; L.C., A.B., J.A.B.L. and E.C.R. designed, prototyped, and fabricated devices; L.C., A.B., K.A.K., E.B.V. and D.A.G. carried out benchtop characterization; L.C., A.B., K.A.K., E.B.V., D.A.G. and D.S.M. carried out animal experiments; A.A. performed deep neural network analysis; R.P. performed mechanical simulation; A.B. and M.J. wrote custom MATLAB and Android software; L.C., A.B., A.A., R.P., D.S.M. and P.G. analyzed data; L.C., A.B., D.S.M. and P.G. wrote the manuscript.

## Competing interests

The authors declare no competing interests.
