## [Peer Review File. · Nature Communications]

Osseosurface electronics – Thin, wireless, battery-free and multimodal musculoskeletal biointerfacesREVIEWER COMMENTS

Reviewer #1 (Remarks to the Author):

The manuscript presented by Cai et al. describes the wireless biomodulation system design to interact with the musculoskeletal system, especially with the bone surface. The Authors term their device "osseosurface electronics" and aim to seamlessly integrate the electrical probe with the bone surface for real-time recording and stimulation of tissues for studies in animal models and future translational therapeutic applications.

The authors first describe the electrical design of the sensor and perform the system characterization under the bench-top conditions, followed by the characterization of the flexible interconnects' mechanical stability. The authors developed two types of devices: for small animals and large animals, where the difference is the length of interconnects between the control electronics and biointerface probe. Proof-of-concept for applications in in vivo studies is demonstrated using a rat model; however, not much of a recording data is shown, and analysis focuses on the biocompatibility and recovery from surgery. The application in large animals is demonstrated by implantation onto the surface of sheep humerus in the in situ model. Finally, the authors present the 3D printed device that would help transfer the device to the bone surface, but the design does not seem to fit the surgical needs.

Conformable, biocompatible, minimally invasive, and wireless implants that enable probing into the musculoskeletal system's functions are important for the field of bioelectronics, and many of the concepts presented in the manuscript would be of interest to the field and broad audience. I would recommend accepting this manuscript after some revisions.

The specific points of concern in the manuscript are:

1. The bench-top characterization of the devices is presented in a confusing way. First, the extensive characterization of power harvesting capabilities of the device, both theoretical and experimental, is described in detail, but the dissemination of critical conclusions is not provided. E.g. what was the essential design parameter that allowed to achieve good power harvesting through the tissues? How were the physiologically relevant heights and depths determined? Second, the bench-top characterization of biointerface in the main text seems rather straightforward, but the Methods section suggests that not all of the device sensors were operated remotely during this characterization, but some were driven using bench-top instruments. Could the Authors explain this design? E.g. Shouldn't it be possible to power up the u-LED and measure thermistor readings simultaneously through the NFC interface instead of driving the u-LED with a function generator. Third, the stretchability of the self-similar interconnects is extensively described in the Fig. 4. However, this aspect of flexible electronics

has been extensively studied in the literature (e.g. <https://doi.org/10.1038/ncomms2553>). I would recommend to minimize this discussion in the manuscript and reassign the panels from Fig. 4 to the previous figures and SI. Also, because the flexible interconnects are part of the device's mechanical design, it would seem more natural to describe this aspect before the biointerface characterization.

2. For biointerface studies, limited data is shown. For the temperature profile in Fig. 5B is the shaded region the uncertainty within one subject or the group of subjects? The same comment applies for the Fig. 5E. In Fig. 5C, only 2 load-unload cycles are shown. More representative traces should be shown to demonstrate the device's recording capabilities. Also, only one frequency of the response is shown. It would be beneficial to show recording capabilities at different stride frequencies in the subject to assess the bandwidth of the recording. It would also be very valuable to demonstrate temperature recording in induced hypo- and hyperthermia experiments and during physical activity. Utilization of u-LED for optogenetic control in vivo would also be helpful.

3. The photo description of rat surgical protocol is valuable in the manuscript. On the other hand, the sheep protocol is a little confusing. The Methods section suggests that the animal was under anesthesia, but the usage of the word "cadaver" in Fig. S15 would indicate that the animal was already euthanized. Could the Authors clarify this aspect?

4. The applicator device shown in Fig. S16 does not seem to be very useful if the surgical incision is to be very small (like the one shown in Fig. S15) and would work only on the wholly exposed bone, which might not be clinically significant.

5. I would suggest refraining from using the terms "TX antenna" and "RX antenna" when writing about NFC communication as the system is used for two-way communication, and each antenna is used for both transmission and receiving. Alternatives such as "initiator antenna" and "target antenna" or other terms can be used to avoid confusion.

Reviewer #2 (Remarks to the Author):

This manuscript reports the design and demonstration of flexible, wireless, and battery-free devices that can interface with non-planar bone surfaces and provide continuous monitoring of musculoskeletal parameters. The devices feature impressive integration of commercial chips in a thin and flexible form factor, and are shown to provide non-debilitating monitoring in rats over 2 weeks as well as operation at the human scale in sheep carcass. These results convincingly demonstrate the potential for flexible

electronics to interface with bone surfaces and wirelessly monitor health, which suggests exciting scientific and clinical applications in orthopedics. I recommend that the paper be published in Nature Communications provided that the comments below are addressed.

1. Results (bottom of page 3): The authors should discuss the design requirements that are specific to osseosurface electronics, as opposed to epidermal electronics for which many of the design concepts used were originally developed. For example, how are the mechanical considerations different, given that bone is non-planar but rigid? How do the footprint requirements change, considering the different depth of operation?

2. Results (page 5) "...reliable power harvesting through thick tissues (> 10 cm)". It is more appropriate to say "up to 8 cm", considering the maximum tested depth in Fig. S17.

3. Fig. 5B. Caption should explain what the pink shaded region represents.

4. Fig. 5C/E. How well do the strain measurements from the device compare to the stride measurements using the camera?

5. Page. 24. Outcomes of the 5 rats used in the study should be discussed. Did any devices fail within the 2 weeks, and if so what were the observed modes of failure?

6. Discussion (page 16). A limitation of current study is that the device was not entirely on the bone surface (due to inherent size constraints) in the chronic rat study (Fig. 4A). What aspects still need to be evaluated in large animals studies should be mentioned here.

7. Methods. Details for the computational electromagnetic design should be provided.

8. Fig. S5. Inductance and resistance of the antenna should be reported.

9. Fig. S14. What specific tissue region around the device is the histology from?

John Ho, Assistant Professor, National University of Singapore

Reviewer #3 (Remarks to the Author):

Le Cai and co-workers here report on a type of flexible, wirelessly powered electronic biointerface patch that can be attached to bone surfaces for chronic implantation, that the authors consequently term "Osseosurface electronics".

The paper is well organized and written, and certainly adds an important piece to the growing field of bioelectronics. However, the device architecture, methods of fabrication, wireless data readout and set of sensors (strain, temperature, and a u-ILED) closely resemble those of previous work from the group and others in the field. While this should not per se be understood as criticism on the work itself, a few points should be made clearer in order for this platform to become more convincing.

As I understood, the main idea of mounting this type of electronics on bone is to increase chronic operational lifetime by reducing mechanical impact associated with animal (and later eventually human) motion on the electronics itself. On page 17 the authors however state that their current method of bonding using cyanoacrylates limits lifetime to weeks ("Currently we expect device lifetimes on the order of several weeks limited by the cyanoacrylate attachment to the bone, an interface that gradually degrade"), taking into account only said interface though not the operational lifetime of the electronics under in vivo conditions itself. In the current manuscript, experimentally supported arguments on this form of electronics being more chronically stable than similar, previously reported forms are moderately convincing for the following points:

1) Many of the bench top cyclic loading experiments are performed for a very limited number of cycles only, a few tens of seconds of operation are shown mostly (i.e. Figure S8 for tests of the bone mounted strain sensors, Figure 3 B,C,H only a few cycles are shown). Maybe I missed it somehow, but it seems especially such bench top experiments with the devices mounted on bones or bone models lend themselves to corroborate long term (mechanical) stability of the devices on such interfaces.

2) There are tests where the devices are submerged in PBS for about 40 h, and continue to transmit data wirelessly. I wonder why only readings of the temperature sensor are shown, but not the ones from strain sensors and eventually a test including the u-ILEDs? I am aware that such tests of device stability are challenging, but a clear advantage of the integration on the bone surface in terms of device stability should be demonstrated in such experiments.

3) Limiting factors on in-vivo lifetime of the devices beyond a failure of the cyanoacrylate bonding should be discussed and eventually evaluated in bench-top experiments. I would assume that issues on bodily fluids penetrating encapsulation layers should arise. Here, it would be interesting to see if the bone-electronics interface would actually have some additional benefits in terms of reduced exposure to body fluids, increasing lifetime? Given the difficulty and time and cost expense of in-vivo studies, I would suggest a more thorough bench top characterization similar to the one demonstrated in Figure 2I, but including all sensor/actuator components of the electronic device.

4) The rodent study summarized in Figure 4 demonstrates that implantation of the devices has little impact on the gait of the rodents for at least two weeks. But it seems it does not demonstrate device functionality beyond 10h (Figure 5B), or did I miss this? Again, the two in vivo recorded strain cycles of Figure 5 C make me wonder if the authors could include more of their data to underline functionality of the implanted platform.

5) Figure 4, the rodent study. I may have missed it, but the u-ILED seems not be involved in the recording of any physiological data or any form of (measured) optical stimulation. If the authors choose to include this part in their Figure, I would recommend to at least demonstrate that the u-ILED remains functional for the period of implantation, opening up the possibility for further studies involving this type of devices for optical bio interfaces.

In summary, the authors present a very interesting electronic platform that interfaces with bone surfaces. However, the current in vivo demonstrations in rodents and ex-vivo experiments with sheep bones or on deceased sheep seem to be not ideal in transporting the idea of having developed a chronically stable biointerface. Here, functionality of the whole multimodal devices is not demonstrated very convincingly. I recommend to mitigate those issues mostly via bench-top experiments involving explanted bones or bone surrogates and PBS. Demonstrating a clear advantage of the bone-electronics interface on device longevity would certainly multiply the impact of this study.

Green – response to the reviewers' comments

Blue – Modification made to the manuscript

Red – Changes made to text

Reviewer #1

General comments:

The manuscript presented by Cai et al. describes the wireless biomodulation system design to interact with the musculoskeletal system, especially with the bone surface. The Authors term their device "osseosurface electronics" and aim to seamlessly integrate the electrical probe with the bone surface for real-time recording and stimulation of tissues for studies in animal models and future translational therapeutic applications.

The authors first describe the electrical design of the sensor and perform the system characterization under the bench-top conditions, followed by the characterization of the flexible interconnects' mechanical stability. The authors developed two types of devices: for small animals and large animals, where the difference is the length of interconnects between the control electronics and biointerface probe. Proof-of-concept for applications in in vivo studies is demonstrated using a rat model; however, not much of a recording data is shown, and analysis focuses on the biocompatibility and recovery from surgery. The application in large animals is demonstrated by implantation onto the surface of sheep humerus in the in situ model. Finally, the authors present the 3D printed device that would help transfer the device to the bone surface, but the design does not seem to fit the surgical needs.

Conformable, biocompatible, minimally invasive, and wireless implants that enable probing into the musculoskeletal system's functions are important for the field of bioelectronics, and many of the concepts presented in the manuscript would be of interest to the field and broad audience. I would recommend accepting this manuscript after some revisions.

Our response: We thank the reviewer for the positive feedback and have addressed the reviewers questions with additional experiments and modifications to the manuscript.

Specific points:

- 1. The bench-top characterization of the devices is presented in a confusing way.**
 - a. First, the extensive characterization of power harvesting capabilities of the device, both theoretical and experimental, is described in detail, but the dissemination of critical conclusions is not provided. E.g. what was the essential design parameter that allowed to achieve good power harvesting through the tissues? How were the physiologically relevant heights and depths determined?**

Our response: We thank the reviewer for the comments and have addressed these questions by additional supplementary figures and additional discussion in the manuscript.

Generally, the advanced energy harvesting capability of the system is primarily attributed to the choice in operation frequency (13.56 MHz) and the near field operation mode. Near field power transfer is efficient and 13.56 MHz operation

frequency has low tissue attenuation, hence the high energy availability at the implant.

More specifically antenna design, both for primary antenna and secondary antenna have been optimized for the application at hand (use in small animal subjects and use in large animal models). An example of this optimization is the primary antenna design which is critical to provide continuous power to the device regardless of subject orientation within a given experimental paradigm. In our case the animal subject is located on a treadmill and the device is located on the back of the subject and will, depending on subject and behavior vary with respect to cage floor.

In our experimental paradigm the subject walks on a treadmill and the implant location is around 3-6 cm from the cage floor. Which we have documented in a supplemental figure. The primary antenna is designed to provide the highest magnetic field strength in this volume and therefore provide sufficient power to the device. Characterization of the antenna is presented in figure 2F which indicating good agreement with power transfer capability and implant location during this behavioral paradigm.

Modification made to the manuscript (page 9, line 163):

“The spatial distribution (Fig. 2F) of the harvested power in the 45 cm x 12 cm cage exhibits sufficient power for stable device operation with primary antenna winding heights at 3 cm and 6 cm chosen to match device body implantation location in the freely moving subject (Fig. S9). harvesting at two physiologically relevant heights, i.e. 3 cm and 6 cm that mark the lower and upper position limits during regular gait. Similar results are obtained for the 26 cm x 33 cm TX antenna primary antenna, revealing a rectified voltage of ~ 2.1 V at a load of 300 Ω (14.7 mW) at the cage center and sufficient power at physiologically relevant heights (Supplementary Information, Fig. S68B and C).”

Addition to Supplementary Information (Figure S9):

Figure S9. Photograph of animal postures and height of device body relative to antenna position. A, Photograph of animal walking on treadmill. B, Photograph of animal during head raising.

- b. **Second, the bench-top characterization of biointerface in the main text seems rather straightforward, but the Methods section suggests that not all of the device sensors were operated remotely during this characterization, but some were driven using bench-top instruments. Could the Authors explain this design? E.g. Shouldn't it be possible to power up the u-LED and measure thermistor readings simultaneously through the NFC interface instead of driving the u-LED with a function generator.**

Our response: We thank the reviewer for the pointing this out. Wherever possible and without inducing significant errors, we used the wireless system to characterize the performance of our biointerfaces. Depending on the experiment we use external instruments to enable gold standard instruments to quantify individual component characteristics. For instance, we have characterized the NTC temperature sensor with a heater coil that is driven wirelessly on the device to carefully control temperature applied to the sensors to characterize resolution and dynamic range. In this experiment we patterned a heater on the backside of the polyimide carrier to provide the best thermal coupling to NTC to reduce thermal coupling error and enable the full encapsulation of the device to submerge the system in PBS to reflect operation in the body. For the measurements with the μ -ILED we did not have the have fine control over current implemented in the wireless device, hence we would not be able to test ultimate fidelity of the thermographic recording capabilities. Characterization of the optoelectronic components, such as output of the μ -ILED, we used standard instruments such as function generator and oscilloscope in order to obtain results with high fidelity, which would require significant amount of efforts in hardware development if the characterization were performed exclusively with wireless devices, e.g. applying and monitoring specific amount of current to the μ -ILED with precise timing. We have added the following content in the Method section to clarify.

Modification made to the manuscript (page 30, line 538):

"The thermistor and μ -ILED were immersed in a PBS bath to mimic the in-vivo environments and to prevent the device from overheating at high optical power. A function generator (Siglent, SDG 1032X) was used to drive power to the μ -ILED to test μ -ILED heating of surrounding tissue using a micro-temperature sensor collocated by the μ -LED with varying voltage (1.85 V, 2 V and 2.12 V), frequency (5 – 40 Hz) and duty cycle (5 – 80%), while the temperature sensor signal was wirelessly read out with a handheld antenna."

Modification made to the manuscript (page 28, line 503):

"The optical power was measured with an integration sphere (OceanOptics FOIS-1). The current consumption of the μ -ILED was measured with an Oscilloscope (Siglent SDS 1202X-E) measuring the voltage across ~~by a customized circuit where~~ a 10 Ω resistor in series with the μ -ILED. ~~and a~~The wireless circuit uses a MOSFET (PMZ130UNE, Nexperia) to drive the μ -ILED with programmed pre-defined frequencies and duty cycles stored in the ~~were connected in series with the μ -ILED and a μ Cmicrocontroller~~ (ATTiny 13A, Microchip Technology). was

~~used to switch the MOSFET on and off with pre-defined frequencies and duty cycles. Total current consumption of the device was measured using a benchtop power supply. The circuit was powered by a DC power supply (1.8 V) while with an Oscilloscope (Siglent SDS 1202X-E) was used to recording the voltages drop across the a 10 Ω shunt resistor from which the current could be calculated.~~

Characterization of the wireless thermography: The wireless temperature sensor was immersed in a water bath whose temperature was varied from 33 °C to 41 °C using a hotplate and monitored by a commercial thermocouple digital thermometer. The sensor signal was wirelessly recorded with a handheld antenna using the Feig reader and ISOStart, and used for sensor calibration.

~~In order to characterize the resolution of the wireless thermographic biointerface, the Cu copper on the bottom side of the Pyralux substrate was laser ablated to form a micro-heater beneath the NTC thermistor (design of the micro-heater is shown in Fig. S9A). The micro-heater was wirelessly powered by the rectified voltage with a MOSFET by the osseosurface device and the on board voltage regulator was used to drive PWM controlled 1.8V with a heater element resistance of 5.36 Ω that was characterized prior with a SMU (SMU, Keithley 2450) enabling a defined heater output., as a switch that was controlled by a The μ Cmicrocontroller with was used to control a programmed sequence of duty cycles varying from 24% to 1220% to control the MOSFET that drives the micro-heater. The fully encapsulated circuit was immersed in a water bath and a handheld antenna was used to wirelessly power the device and retrieve temperature recordings from the NTC the sensor signal. The resolution of the temperature sensor was subsequently determined by the smallest sensor response that could be distinguished from the background noise.”~~

- c. **Third, the stretchability of the self-similar interconnects is extensively described in the Fig. 4. However, this aspect of flexible electronics has been extensively studied in the literature (e.g. <https://doi.org/10.1038/ncomms2553>). I would recommend to minimize this discussion in the manuscript and reassign the panels from Fig. 4 to the previous figures and SI. Also, because the flexible interconnects are part of the device's mechanical design, it would seem more natural to describe this aspect before the biointerface characterization.**

Our response: We agree that self-similar interconnects are well explored in literature for use on skin and other bioelectronic applications. However, for our application the stretch ability is only one of the concerns. Impact on the sensor itself, specifically the strain sensor, is important. In other words, we are interested in how much strain we apply to the bone mounted sensing element during behavior of the animal because it can impact biointerface stability and sensor readings. So, for our application we have engineered the self-similar serpentine in a way that strain on the bone strain sensor is minimal and have calculated impact on the sensor using finite element simulations. We agree that this context is not very well communicated in our manuscript, and we have made changes to reflect this thought process and resulting engineering choices and characterizations.

Modification made to the manuscript (page 15, line 242):

“Mechanical characterization. Subdermal implantation in small animal models involves placement in highly mobile areas which require a **compact formfactor with mechanical design strategies** that can withstand repeated strain cycles **without interfering with biosensor readouts**. This is evident when studying the micro-CT scan (Fig. 4A) of a rat with an implanted osseosurface electronic device, where the device body is anchored subdermally around the lumbar vertebra and connection to the biointerface, which is located on the left femur, is accomplished via self-similar serpentine interconnects³⁵.”

Modification made to the manuscript (page 15, line 248):

“The durability of the serpentine interconnect is critically important for reliable in-vivo operation. FEA simulation guided design enables reliable performance under repeated strains of over 250% while maintaining a maximum strain of ~ 0.7% in the ~~Cu~~copper traces which is below 1%, the failure strain of ~~Cu~~copper(Fig. 4B)³⁵. Chronic electro-mechanical stability is confirmed by cyclic straining to 250% for 10,000 cycles, with negligible change in conductivity (Fig. 4C). Critical to the stability of sensor readings during behavior is the mechanical isolation of the stretchable interconnect and the strain sensor, this includes introduction of strain to the bone during deformation of the serpentine interconnect. The biointerface is mechanically isolated by the serpentine interconnects designed to transfer minimal strain during deformation to the bone and the strain sensor biointerface, as confirmed by FEA in Fig. 4D (Fig. S15) showing that stretching the serpentine by 250% does not influence the sensitivity or accuracy of the strain gauge (Fig. 4E-F and Fig. S16, details in Method section), ~~which is further. Further evaluation validated~~ of serpentine electro-mechanics are performed with ~~by~~ bench-top experiments using a servo-hydraulic materials testing system (MTS) indicating stable operations of recording capabilities under multiple cycles of high strains ~~tests~~(Fig. S17) above physiological strains as means to evaluate robustness of the approach. Both benchtop electro-mechanical testing of the monolithic serpentine structure with FEA simulations of the serpentine shows solid performance in strain isolation enabling flexible sensor placement to distal regions with minimal effect of sensor readouts.”

Addition to Supplementary Information (Figure S17):

Figure S17. Extended servo-hydraulic materials testing. A, 10,000 cycle testing on an MTS (Series 810, MTS Systems Corporation). B, First 10 cycles of testing procedure. C, Last 10 cycles of testing.

2. For biointerface studies, limited data is shown. For the temperature profile in Fig. 5B is the shaded region the uncertainty within one subject or the group of subjects? The same comment applies for the Fig. 5E. In Fig. 5C, only 2 load-unload cycles are shown. More representative traces should be shown to demonstrate the device's recording capabilities. Also, only one frequency of the response is shown. It would be beneficial to show recording capabilities at different stride frequencies in the subject to assess the bandwidth of the recording. It would also be very valuable to demonstrate temperature recording in induced hypo- and hyperthermia experiments and during physical activity. Utilization of u-LED for optogenetic control in vivo would also be helpful.

Our response: We thank the reviewer for the valuable suggestion and have undertaken significant new experiments to demonstrate device capabilities extensively.

Firstly we have performed new bench top experiments to demonstrate bandwidth of the system. Here we record increasing bone deformation up to the physical limit of our strain testing apparatus. Our osseosurface electronics provide 87 Hz sampling rates which outperforms gold standard strain gauge readout electronics. This is documented in supplemental figure S12A-D and new panels in figure 5. We were able to record deformation frequencies of 10 Hz in these bench top experiments.

Additional to these bench top results we have performed new animal experiments that show real time readouts of osseosurface electronics of a rat subject walking on the treadmill with time synced strain readouts. We show the measurement in a new supplemental video and discuss results from both experiments in the manuscript.

We have also clarified how thermal experiments were conducted, experiments with hypo and hyperthermia were unfortunately not possible under our current experimental protocol. The most obvious delta occurs during surgery which is why we chose to represent this here in the manuscript.

Additionally, we show optical stimulus capability in the freely moving subject in new supplemental figures.

In previous work we have presented optogenetic stimulation capability in the periphery and have therefore opted to only show light delivery capability^{R1}. However, we are very excited by the possibility to perform optogenetics in the bone (there are neurons in the bone) and are currently planning experiments that will be subject of future studies.

R1. Gutruf, P. *et al.* Wireless, battery-free, fully implantable multimodal and multisite pacemakers for applications in small animal models. *Nat. Commun.* **10**, 5742 (2019).

Modification made to the manuscript (page 11, line 200):

“Wireless strain sensors exhibit sensing performance on par with the gold standard wired strain measurement systems in typical physiologically relevant range (0-1200 $\mu\epsilon$)²², as confirmed by ramping and cyclic loading tests performed on a sheep bone specimen (Fig. 3A-C, details in Methods section), with an estimated sensitivity of ~ 3.5 ADC/ $\mu\epsilon$ and a resolution of ~ 14.3 $\mu\epsilon$ with a sampling rate up to 87Hz. Cyclic loading tests up to 10 Hz show recording capabilities that surpass gold standard acquisition systems for recording bone strain (Fig. S12A-C). Simultaneous recording of temperature and strain with optogenetic stimulation is shown in supplementary figure S12D. Repeatable, hysteresis-free response is obtained upon progressively increasing loads (Fig. 3B), indicating that a stable bond is formed between the strain gauge and the bone specimen.”

Addition to Supplementary Information (Figure S12):

Figure S12. Electromechanical benchtop testing of wireless device on sheep femur. A, 1 Hz sine load cycle. B, 5 Hz sine load cycle. C, 10 Hz sine load cycle. D, 1 Hz triangle load cycle with active convective heating is heat gun of the bone and device to match average body temperatures in a rat.

Modification made to the manuscript (page 17, line 281):

“The optoelectronic interface attached on the bone surface allows for direct optical stimulation of skeletal muscles in freely-moving subjects, as shown in Fig. 5A where illumination of the muscle is visually validated, highlighting a systematic advantage

over traditional transdermal light sources³⁰ or optical fiber-based techniques²⁹ which are challenging to implement in highly mobile areas. Light delivery can be programmed to control frequency and duty cycle of stimulation each with 16-bit precision. Chronic functionality is demonstrated with in frame analysis of tracked red pixel intensity over time (Fig. S19) from video recordings 22 days post implantation (Supplementary Video V1). In-vivo device operation of temperature sensing capabilities is tested over extended time periods is—demonstrated by a 10-minute moving window of thermographic recording that show recovery of the animal post-surgery for over 11 hours in the home cage (Blue) with a standard deviation within the 10-minute moving window (pink) (Fig. 5B). Following the surgical implantation of the device a dip in local body temperature during the first half hour of recovery after implantation is observed which is an expected hypothermia hypothermic response to isoflurane anesthesia³⁶ and transfer from a warmed surgical table to a cooler recovery area.”

Addition to Supplementary Information (Figure S19):

Figure S19. Tracked red light intensity of optogenetic stimulation while implanted with control of frequency and duty cycle: 9 Hz, 50% (red); 9 Hz, 100% (green); Off (yellow); 5 Hz, 75% (blue).

Addition to Supplementary Information (Video V1):

Video V1. Video of light delivery with biointerface implantation location on the rat femur.

Modification made to the manuscript (page 18, line 299):

“Strain profile recorded wirelessly in a custom treadmill arena with the 45 cm x 12 cm ~~TX antenna~~ primary antenna reveals characteristic loading and unloading phases with absolute strain values comparable to literature values for the rat femur obtained by tethered sensors²².

Time synced wireless recording of strain and temperature are compared with video recording of the animal during walking periods after a week of recovery (Supplementary Video V2). Analysis enables intimate insight into bone strain of the femur during gait. The recordings of strain show four distinct points during a gait cycle: 1) Mid-stance, 2) Lift-off, 3) Mid-swing, 4) Touch-down^{37,38}. These points and their corresponding frames are shown in Figure 5C. Behavioral assays, gait analysis in particular, are useful tools to investigate pathogenesis and develop new therapeutics for musculoskeletal disorders such as osteoarthritis³⁹.”

Additions made to References (Ref 34, Ref 35):

- 37. Fujiki, S. et al. Adaptive hindlimb split-belt treadmill walking in rats by controlling basic muscle activation patterns via phase resetting. Sci. Rep. 8, 17341 (2018).*
- 38. Dienes, J. A. et al. Analysis and Modeling of Rat Gait Biomechanical Deficits in Response to Volumetric Muscle Loss Injury. Front. Bioeng. Biotechnol. 7, 146 (2019).*

Modification made to the Figure (Figure 5):

Figure 5. In-vivo studies of OSE in rats. **A.** Photograph of the rear part of a rat featuring the μ -LED attached on the femur illuminating through muscle and skin. **B.** Temperature profile recorded in-vivo during the period of ~ 11.5 hours following the implantation surgery. Inset, schematic of the rat residing in the home cage. **C** Time synchronized strain recording and corresponding video frames of the rat gait cycle. ~~Strain profile recorded in-vivo while the rat's rear leg is being pulled manually.~~ **D-E.** Comparison of the rat's gaits at different stages of the study: before surgery, 1 week after surgery and 2 weeks after surgery: photograph of rats walking on the treadmill overlaid with the trajectory of the ankle (**D**); key parameters – stride frequency, stride length and duty factor – that characterizes the rat's gaits (**E**). Color of the dots in **D** represents the velocity of the ankle.”

Addition of Supplementary Information (Video V2):

Video V2. Video with corresponding plot of wireless recording of strain of the left femur during gait.

Modification made to the manuscript (page 17, line 289):

“In-vivo device operation of temperature sensing capabilities is tested over extended time periods is demonstrated by a 10-minute moving window of thermographic recording that show recovery of the animal post-surgery for over 11 hours in the home cage (Blue) with a standard deviation within the 10-minute moving window (pink) (Fig. 5B). Following the surgical implantation of the device a dip in local body temperature during the first half hour of recovery after implantation is observed which is an expected hypothermia hypothermic response to isoflurane anesthesia³⁶ and transfer from a warmed surgical table to a cooler recovery area.”

Modification made to the manuscript (page 13, line 213):

“To demonstrate the feasibility of optical-stimulation-optogenetic stimulation capability in the periphery³³ on an osseosurface electronic platform, we use miniaturized individually addressable μ -ILEDs to deliver stimulation to the bone and the surrounding tissue. The bone-tissue interface provides a unique platform capable of versatile optical coupling modes tailored for various application scenarios, such as phototherapeutic stimulation for bone regeneration²⁸ and optogenetic activation of muscular contraction³². Fig. 3D demonstrates this design flexibility with devices that are capable of illuminating the soft tissue side (left), bone side (middle), and multimodal stimulation (right). Typical operation for optical stimulation covers a parameter space of 5 - 20 Hz for applications such as sustained tetanic contraction^{31,32}, which is achieved on our platform by utilizing a μ Cmicrocontroller for precisely controlled timing as demonstrated in figure 3E Fig-3E.”

Additions made to References (Ref 33):

33. Gutruf, P. et al. Wireless, battery-free, fully implantable multimodal and multisite pacemakers for applications in small animal models. Nat. Commun. 10, 5742 (2019).

3. The photo description of rat surgical protocol is valuable in the manuscript. On the other hand, the sheep protocol is a little confusing. The Methods section suggests that the animal was under anesthesia, but the usage of the word "cadaver" in Fig. S24 would indicate that the animal was already euthanized. Could the Authors clarify this aspect?

Our response: We agree with the reviewer that this is not clearly stated in the manuscript. The sheep was euthanized right before implanting our devices, a requirement of the clinical protocol. This is why we could measure the cooling of the cadaver. We have added more information to the manuscript to clarify the experimental procedure.

Modification made to the manuscript (page 21, line 353):

*"In-situ studies in large animal models. The scalability of our platform enables the use in large animal models with minimal modifications. **Immediately after euthanization (16-202 IACUC), in-situ operation of devices designed for large animals are demonstrated on sheep humeri where small footprint and soft mechanics of the device enable conformal application to the bone surface (Fig. 6Fig. 7A and Fig. S24A-B-Fig. S15A-B).**"*

4. The applicator device shown in Fig. S25 does not seem to be very useful if the surgical incision is to be very small (like the one shown in Fig. S24) and would work only on the wholly exposed bone, which might not be clinically significant.

Our response: We thank the reviewer for pointing this out. We envision the implantation of the device in its current form during surgeries that require an opening of the tissue around the bone, such as fracture repair and placement of artificial bone replacements. One of the most challenging aspects of device placement is the uniform application to the bone because of the need of excellent contact to enable strain transfer to the biointerface. We designed the applicator to enable application of superglue to the device and subsequent one step lamination to the bone. For future revisions of this device and options that involve bone bonding using CPC particles as outlined in new experiments in response to reviewer 3 comment 3 more minimally invasive designs are possible. To clarify applicator design choice, usage and future devices utilizing CPC bone bonding we have added a discussion and additional descriptions to the manuscript.

Modification made to the manuscript (page 19, line 320):

"The welfare of the test animals is also reflected by the steady weight-gain from two days post-surgery until the study is terminated (Fig. S20Fig. S13). In addition, histological analysis shows that the **subcutaneous implantation body of the device body in the lumbar vertebra region**, is surrounded with fibrous tissue containing blood vessels without any evidence of inflammation or significant foreign body response after two weeks of in-vivo operation (Fig. S21A-E).

Permanently attached Osseosurface electronics. The ultrathin electronic platform enables direct lamination onto the bone which provides the opportunity to attach devices permanently to gather information on bone health long term. Because of cell

turnover any glue will exhibit limited lifetime and continuously degrading biointerface quality. A solution to this problem is to directly grow osseosurface devices to the bone using of CPC particles⁴¹. For the use in animal subjects fast adhesion is critical to enable accelerated experimental timelines, which can be accomplished by the addition of transforming growth factor beta 1 (TGF- β 1), an osteogenic protein that enables rapid bone bonding to the CPC particles⁴². We use this scheme to bond CPC particles to the osseosurface electronics using implant grade epoxy, and subsequently applying TGF- β 1 to the CPC particles. The resulting layered device composition is shown in figure 6A. The exposed particles enable effective bone bonding after implantation and temporary fixation with resorbable sutures (Fig. 6B). The efficacy of this approach can be observed in vivo by recording the average delta strain values of gait on a treadmill. Figure 6C shows the evolution of the delta strain over the course of several weeks, here we can observe weak bonding of the strain gauge to the bone for the day of the surgery with rapid increase in delta strain values in the following days and a stable attachment after a week (Fig. S22). The successful growth of the device to the bone can also be observed after explantation (day 27) and cross-section preparation and subsequent secondary electron imaging (Fig.6D and Fig. S23), details in methods section. The micrograph shows successful growth of the bone to the particles that permanently affix the flexible electronics enabling chronic recording of bone health. The chronic application of these devices enables a multitude of new studies for example strain mediated bone remodeling with experimental paradigms that involve extensive mobility of subjects and longitudinal studies with multiple co-habiting subjects which is facilitated by complete recovery after implantation of the wireless and battery free device as shown in Fig.6E.

In-situ studies in large animal models. The scalability of our platform enables the use in large animal models with minimal modifications.”

Additions made to References (Ref 36):

41. Szivek, J. A., Anderson, P. L., Dishongh, T. J. & DeYoung, D. W. Evaluation of factors affecting bonding rate of calcium phosphate ceramic coatings for in vivo strain gauge attachment. *J. Biomed. Mater. Res.* 33, 121–132 (1996).
42. Cordaro, N. M., Szivek, J. A. & DeYoung, D. W. Surface enhancements accelerate bone bonding to CPC-coated strain gauges. *J. Biomed. Mater. Res.* 56, 109–119 (2001).

Modification made to the manuscript (Page 35, Line 647):

"Finally, the applicator was carefully peeled off with minimal out-of-plane sheer force between the membrane, device encapsulation, and adhesive agent.

Calcium phosphate ceramic coating

Implant-grade epoxy (Master Bond EP42HT, Master Bond, Inc., Hackensack, NJ) was prepared according to the manufacturer's instructions and a thin even layer was blade coated to the bottom surface of the strain gauge (N2A-06-S5182N-10C/E4, Micro-Measurements). Spherical crystalline calcium phosphate ceramic (CPC2, CeraMed) was added to cover the surface of the epoxy and silicone rubber was used to gently apply downwards pressure on the CPC particles while curing the epoxy for 24 hours at

room temperature. All CPC coatings were inspected for surface exposure without epoxy coating to enable bond to the bone. Devices were coated with TGF- β 1 to accelerate bone to CPC bonding.

Scanning electron microscope imaging of CPC – bone adhesion

A rat after 27 days post-implantation was euthanized, and the femur was explanted along with the attached strain gauge. The bone was cleaned, removing any soft tissue and fixed using a neutral buffered formalin for 2 hours. The bone was dehydrated in a solution of ethanol (EtOH) starting at 70% for 2 hours and increasing concentrations for at least 2 hours as follows: 80% EtOH, 95% EtOH, 95% EtOH, 100% EtOH, 100% EtOH, xylenes, 100% EtOH. The bone was embedded in methyl methacrylate (Technovit 9100 kit, Kulzer) through a pre-infiltration process for 2 hours at room temperature and an infiltration process within a vacuum chamber for 15 minutes to remove any bubbles. The embedded bone is then cured at -4C over night. The bone and strain gauge were cut exposing the cross-sectional view. The cross-section surface was smoothed and polished using a grinding wheel (GP-25, Leco) from 120 grit to 600 grit. The surface was gold sputtered (Hummer 6.3, Anatech USA) allowing for a gold thickness of 9 nm. SEM images were taken with various magnifications (60x, 120x, 280x, 300x, 600x) around the bone strain gauge interface using (Inspect S50, FEI) operating at 30kV.

Device operation in deep tissue

Slits of ~ 3 cm length were cut into the porcine tissues and ~~(2 pieces layers of tissue were stacked to increase device – primary antenna distance, as shown in 4 cm thick each, Fig. S4926 for a test at 8 cm device- primary antenna distance) at various thicknesses.~~"

Modification made to the Figure (Figure 6):

Figure 6. Promotion of osteogenesis using surface-engineered calcium phosphate ceramic coatings. A. Photograph of the bottom coated with CPC and a

side view diagram of layer composition of the strain gauge sensor with CPC. **B.** Image of device 3 weeks post implantation. **C.** Wireless strain gauge data collected over 2 weeks post implantation showing stable CPC adhesion. **D.** Single electron microscope image of cross section of CPC adhesion to bone. **E.** Animal healing after implantation.

Addition of Supplementary Information (Fig. S23):

Figure S23. Additional SEM cross section images of CPC bone bonding. A. SEM cross section of strain gauge bonded to the bone using CPC particles with a magnification of 120x and a photograph image inset. **B.** SEM cross section with a magnification of 280x, **C.** 300x, and **D.** 600x.

Modification made to the manuscript (page 24, line 400):

"This offers unprecedented opportunities for mechanistic studies of osteogenesis and pathogenesis of musculoskeletal diseases, as well as the development of new types of diagnostics and therapeutics. Furthermore we demonstrate the successful attachment of the Osseosurface electronic biointerface to the bone utilizing surface engineered CPC particles, an important aspect of this technology that enables recording on the order of many years enabling device operation for the lifetime of the subjects without requiring secondary surgery. ~~Currently we expect device lifetimes on the order of several weeks limited by the cyanoacrylate attachment to the bone, an interface that gradually degrade 41,42. Future development can enable the formation of a permanent interface with the bone by exploring bioactive coatings that facilitate direct integration with the osseosurface, allowing device operation for the lifetime of the subject without requiring secondary surgery.~~

This chronic platform can then enable acquisition of holistic data of bone health status and closed loop therapeutic intervention to facilitate treatment and rehabilitation."

Modification made to the abstract (page 2, line 36):

"Battery-free device architecture, *direct growth to the bone via surface engineered*

calcium phosphate ceramic (CPC) particles, demonstration of operation in deep tissue in large animal models and readout with a smartphone highlight suitable characteristics for exploratory research and utility as a diagnostic and therapeutic platform."

Modification made to the abstract (page 32, line 588):

The strain gauge was fixed to the femur using a cyanoacrylate adhesive (M-Bond 200, Micro-Measurements). *For implantation of CPC coated devices in small animals, a 5-0 vicryl suture lassoed around the biosensor was used to pass the sensor from the lumbar region through the subcutaneous tunnel to the thigh. This lasso technique was used to avoid manipulation of the device with sharp surgical instruments and minimize the risk of damage to the encapsulating layers. CPC coated gauges were secured to bone using two 5-0 vicryl sutures without adhesive.* Layered closure was performed using 5-0 vicryl for fascia and 4-0 Quill for subcutaneous tissues prior to recovery from anesthesia.

Modification made to the manuscript (page 34, line 638):

"Osseosurface electronics bone attachment

An applicator comprised of a 3D-printed frame and an elastomeric membrane (~~details in Fig. S25Fig-S16~~) was utilized to facilitate fast attachment of the device on the bone surface. Before attaching the device, the surface of the bone was abraded. The device was first attached to the applicator with the top side in contact with the elastomeric membrane. Adhesive agent (M-Bond 200, Micro-Measurements) was then applied to the bottom side of the device. *The applicator was designed to make surgical process in large animals easier by improving ability to conformally mount the strain gauge to the bone in one step. The applicator was subsequently placed on the bone and pressed gently against the bone in order to conform the device to the bone surface. Next, Once placed,* the device was *then* firmly pressed against the bone with an even force until the adhesive agent cured. Finally, the applicator was carefully peeled off with minimal out-of-plane shear force between the membrane, *device encapsulation, and adhesive agent."*

5. I would suggest refraining from using the terms "TX antenna" and "RX antenna" when writing about NFC communication as the system is used for two-way communication, and each antenna is used for both transmission and receiving. Alternatives such as "initiator antenna" and "target antenna" or other terms can be used to avoid confusion.

Our response: We thank the reviewer for the constructive inputs. We replaced TX antenna and RX antenna to primary antenna and secondary antenna, with the readout antenna as the primary antenna.

Modification made to the manuscript (page 6, line 109):

"Near-field magnetic resonant coupling (13.56 MHz, specific absorption rate (SAR) < 20 mW/kg¹²) between an external primary loop antenna (~~TX-antenna primary antenna~~) and the on-board loop antenna (~~RX-antenna-secondary antenna~~) enables, for the first

time, reliable power harvesting through thick tissues (> 10 cm) with hardware that is compatible with NFC protocols widely available in portable devices¹⁷.”

Modification made to the manuscript (page 6, line 111 | page 8, line 144 & 146 | page 9, line 169 | page 11, line 184 | page 18, line 300 | page 26, line 449 & 451 & 453 | page 27, line 475 & 478 | page 31, line 551 | page 33, line 600 & 602):

~~TX-antenna primary antenna~~

Modification made to Figure Captions (Figure 2 | Figure 5 | Figure S6):

~~TX-antenna primary antenna~~

Modification made to the manuscript (page 6, line 111 | page 7, line 136 | page 8, line 153):

~~RX-antenna-secondary antenna~~

Modification made to the manuscript (page 8, line 147):

(2 - 8 W of ~~RFRX~~ power)

Modification made to Figure Captions (Figure S4 | Figure S5 | Figure S6):

~~TX-antenna secondary antenna~~

Reviewer #2 (Remarks to the Author):

General comments:

This manuscript reports the design and demonstration of flexible, wireless, and battery-free devices that can interface with non-planar bone surfaces and provide continuous monitoring of musculoskeletal parameters. The devices feature impressive integration of commercial chips in a thin and flexible form factor, and are shown to provide non-debilitating monitoring in rats over 2 weeks as well as operation at the human scale in sheep carcass. These results convincingly demonstrate the potential for flexible electronics to interface with bone surfaces and wirelessly monitor health, which suggests exciting scientific and clinical applications in orthopedics. I recommend that the paper be published in Nature Communications provided that the comments below are addressed.

Our response: We thank the reviewer for the positive evaluation of the manuscript and the insightful comments that we used to improve the quality of the work.

Specific points:

- 1. Results (bottom of page 3):** The authors should discuss the design requirements that are specific to osseosurface electronics, as opposed to epidermal electronics for which many of the design concepts used were originally developed. For example, how are the mechanical considerations different, given that bone is non-planar but rigid? How do the footprint requirements change, considering the different depth of operation?

Our response: We thank the reviewer for the constructive suggestion. We have added discussion in the manuscript to highlight additional consideration that are required for osseosurface electronics. An example are the additions in response to reviewer 1 comment 1c which discusses requirements for self-similar interconnects, originally designed for stretchable epidermal electronics. In the context of osseosurface electronics they require special attention to reduce the strain applied onto the biointerface to minimize induced noise during behaviors. In the context of the devices directly applied to the bone in large animal models serpentes do not actually see regular and repetitive strain because the device is completely adhered to the bone. Here serpentes are used to place the bio interface during surgery to locate the device body conveniently on the bone and place the biointerface precisely at the target location. After this placement the device is permanently adhered to the bone. For footprint requirements we have added a discussion of device size vs space at the target location and use in several experimental paradigms.

Modification made to the manuscript (page 4, line 80):

“Device design. The creation of osseosurface electronics requires several technical innovations that differ from epidermal electronics such as ~~to enable a~~ device footprint and mechanical properties suitable for direct lamination onto the bone surface to minimize mechanical mismatch with the surrounding tissues, and electromagnetic design allowing for direct readout through thick tissues with portable devices to enable smart therapeutics. Special attention to mechanical design of interconnects is required

to enable chronic stability of the interconnect and minimize mechanical impact on the targets sensing region to avoid introduction of additional strain. Figure 1B shows a device that meets these design criteria (2.5 cm x 1.5 cm, ~ 170 mg) and enables direct, conformal lamination to the curved osseosurface with minimal impact on the surrounding tissues.”

Modification made to the manuscript (page 15, line 242):

*“**Mechanical characterization.** Subdermal implantation in small animal models involves placement in highly mobile areas which require a **compact formfactor with mechanical design strategies** that can withstand repeated strain cycles **without interfering with biosensor readouts**. This is evident when studying the micro-CT scan (Fig. 4A) of a rat with an implanted osseosurface electronic device, where the device body is anchored subdermally around the lumbar vertebra and connection to the biointerface, which is located on the left femur, is accomplished via self-similar serpentine interconnects³⁵.”*

Modification made to the manuscript (page 15, line 248):

*“The durability of the serpentine interconnect is critically important for reliable in-vivo operation. FEA simulation guided design enables reliable performance under repeated strains of over 250% while maintaining a maximum strain of ~ 0.7% in the **Copper** traces which is below 1%, the failure strain of **Copper**(Fig. 4B)³⁵. Chronic **electro-mechanical** stability is confirmed by cyclic straining to 250% for 10,000 cycles, with negligible change in conductivity (Fig. 4C). **Critical to the stability of sensor readings during behavior is the mechanical isolation of the stretchable interconnect and the strain sensor, this includes introduction of strain to the bone during deformation of the serpentine interconnect.** The biointerface is mechanically isolated by the serpentine interconnects **designed to transfer minimal strain during deformation to the bone and the strain sensor biointerface**, as confirmed by FEA in Fig. 4D (Fig. S15) showing that stretching the serpentine by 250% does not influence the sensitivity or accuracy of the strain gauge (Fig. 4E-F and Fig. S16, details in Method section), ~~which is further.~~ **Further evaluation validated of serpentine electro-mechanics are performed with bench-top experiments using a servo-hydraulic materials testing system (MTS) indicating stable operations of recording capabilities under multiple cycles of high strains tests (Fig. S17) above physiological strains as means to evaluate robustness of the approach. Both benchtop electro-mechanical testing of the monolithic serpentine structure with FEA simulations of the serpentine shows solid performance in strain isolation enabling flexible sensor placement to distal regions with minimal effect of sensor readouts.***

***In-vivo studies in rodents.** The subdermally implantable form factor and wireless, battery-free operation enable an in-vivo multimodal bi-directional interface with the musculoskeletal system without compromising the free motion of subjects in various test arenas (Fig. 5).”*

Addition to Supplementary Information (Figure S12):

Figure S12. Electromechanical benchtop testing of wireless device on sheep femur. A, 1 Hz sine load cycle. B, 5 Hz sine load cycle. C, 10 Hz sine load cycle. D, 1 Hz triangle load cycle with active convective heating is heat gun of the bone and device to match average body temperatures in a rat.

2. Results (page 5) "...reliable power harvesting through thick tissues (> 10 cm)". It is more appropriate to say "up to 8 cm", considering the maximum tested depth in Fig. S1726.

Our response: We thank the reviewer for pointing this out. In Fig. 7F, we showed the measured data up to a thickness of 11.5 cm even though the photographic inset shows the case of 8 cm. We apologize for the confusion. We have made corresponding correction to the manuscript.

Modification made to the manuscript (page 6, line 107):

~~"Fig-Figure 1E describes the electrical working principle of the system that enables wireless, battery-free operation in a form factor suitable for full implantation that can adopt various NFC chipsets (Fig. S3). Near-field magnetic resonant coupling (13.56 MHz, specific absorption rate (SAR) < 20 mW/kg¹²) between an external primary loop antenna (TX-antenna primary antenna) and the on-board loop antenna (RX-antenna secondary antenna) enables, for the first time, reliable power harvesting through thick tissues (>10-cm)-up to 11.5 cm with hardware that is compatible with NFC protocols widely available in portable devices¹⁷."~~

Modification made to the manuscript (page 36, line 674):

"Device operation in deep tissue

Slits of ~ 3 cm length were cut into the porcine tissues and ~~(2-pieces layers of tissue were stacked to increase device – primary antenna distance, as shown in 4-cm-thick each, Fig. S1926 for a test at 8 cm device- primary antenna distance)-at various thicknesses~~. The rectified voltage and data rate were recorded with a handheld primary antenna while the device was loaded with 1 kΩ and inserted into the slits to result in ~~device- primary antenna distances from 0 cm to 11.5 cm~~.

Real-time data visualization through tissues: A device was attached on a sheep's humerus following the device attachment procedures described above, and subsequently covered by a piece of porcine skin (~ 1 cm thick)."

3. Fig. 5B. Caption should explain what the pink shaded region represents.

Our response: We thank the reviewer for pointing this out and have made changes to the manuscript to clarify.

Modification made to the manuscript (page 17, line 289):

"In-vivo device operation ~~of temperature sensing capabilities is tested~~ over extended time periods ~~is-demonstrated by a 10-minute moving window of~~ thermographic recording ~~that show recovery of the animal post-surgery for over 11 hours in the home cage (Blue) with a standard deviation within the 10-minute moving window (pink) (Fig. 5B)~~. ~~f~~Following the surgical implantation of the device ~~a-A dip in local body temperature during the first half hour of recovery after implantation is observed which is an expected hypothermia hypothermic~~ response to isoflurane anesthesia³⁶ and transfer from a warmed surgical table to a cooler recovery area."

4. Fig. 5C/E. How well do the strain measurements from the device compare to the stride measurements using the camera?

Our response: We thank the reviewer for this comment and have addressed this comment with new experiments that show real time experiments in freely moving rats on the treadmill that show direct correlation of the animal motion and strain measured from the osseosurface electronic device. We have added a new supplemental video and have added new experimental readouts to the main figure set.

Modification made to the manuscript (page 18, line 299):

“Strain profile recorded wirelessly in a custom treadmill arena with the 45 cm x 12 cm ~~TX-antenna~~ primary antenna reveals characteristic loading and unloading phases with absolute strain values comparable to literature values for the rat femur obtained by tethered sensors²². Time synced wireless recording of strain and temperature are compared with video recording of the animal during walking periods after a week of recovery (Supplementary Video V2). Analysis enables intimate insight into bone strain of the femur during gait. The recordings of strain show four distinct points during a gait cycle: 1) Mid-stance, 2) Lift-off, 3) Mid-swing, 4) Touch-down^{37,38}. These points and their corresponding frames are shown in Figure 5C. Behavioral assays, gait analysis in particular, are useful tools to investigate pathogenesis and develop new therapeutics for musculoskeletal disorders such as osteoarthritis³⁹.”

Additions made to References (Ref 34, Ref 35):

37. Fujiki, S. et al. Adaptive hindlimb split-belt treadmill walking in rats by controlling basic muscle activation patterns via phase resetting. Sci. Rep. 8, 17341 (2018).

38. Dienes, J. A. et al. Analysis and Modeling of Rat Gait Biomechanical Deficits in Response to Volumetric Muscle Loss Injury. Front. Bioeng. Biotechnol. 7, 146 (2019).

Modification made to the Figure (Figure 5):

Figure 5. In-vivo studies of OSE in rats. **A.** Photograph of the rear part of a rat featuring the μ -ILED attached on the femur illuminating through muscle and skin. **B.** Temperature profile recorded in-vivo during the period of ~ 11.5 hours following the implantation surgery. Inset, schematic of the rat residing in the home cage. **C** Time synchronized strain recording and video frames with matched indication markers for the 4 distinct points during a single cycle of locomotion. ~~Strain profile recorded in-vivo while the rat's rear leg is being pulled manually.~~ **D-E.** Comparison of the rat's gaits at different stages of the study: before surgery, 1 week after surgery and 2 weeks after surgery: photograph of rats walking on the treadmill overlaid with the trajectory of the ankle (**D**); key parameters – stride frequency, stride length and duty factor – that characterizes the rat's gaits (**E**). Color of the dots in **D** represents the velocity of the ankle.

Addition to Supplementary Information (Video V2):

Video V2. Video with corresponding plot of wireless recording of strain in the left femur during an animal walking event.

5. Page. 24. Outcomes of the 5 rats used in the study should be discussed. Did any devices fail within the 2 weeks, and if so what were the observed modes of failure?

Our response: We have now listed all animals implanted with devices including surgery training. We have now added more animals to address reviewer comments and have provided a detailed account of failures and added this to a new supplemental figure. Most common early termination of experiments were related to surgery complications due to initially challenging implantations. This success rate has increased with more device implantations. Devices survived in the animal over 2 weeks with some devices still being implanted in the subject and functional. We have also included extensive bench top accelerated rate tests that investigate device encapsulation predicting device lifetimes of many months.

Modification made to the manuscript (page 17, line 273):

“The implantation of osseosurface electronics in rats involves a skin incision on the back of the subject, device placement into the subcutaneous space, subcutaneously tunneling the biointerface to the limb, attaching the biointerface to the femur with cyanoacrylate, and closing the skin with resorbable sutures (Details provided in the Methods section and Fig. S1218). Devices are tracked and monitored after surgery to gain insight into failure mechanisms. Majority of device failures examined after explanation indicate defects in the Parylene-C encapsulation induced by surgical tools (Supplementary Table 1). Mitigation of these issues in later experiments utilized sutures to manipulate and position the biosensor, resulting in lower failure rates. The optoelectronic interface attached on the bone surface allows for direct optical

stimulation of skeletal muscles in freely-moving subjects, as shown in Fig. 5A where illumination of the muscle is visually validated, highlighting a systematic advantage over traditional transdermal light sources³⁰ or optical fiber-based techniques²⁹ which are challenging to implement in highly mobile areas.”

Addition to Supplementary Information (Figure S11):

Figure S11. Accelerated rate testing at 90°C, 60°C, and 40°C in PBS, testing for wireless communication function, optical stimulation function, standing wave ratio of antenna (Red, Orange, Yellow), and wireless recording of temperature (Navy, Light blue, Blue).

Addition to Supplementary Information (Table S1):

Table S1. Record of device implant and failure mode analysis.

Device	Implantation Period (days)	Device Failure Mechanism
1	14	Data recording was inconsistent with unknown device failure. (Unknown failure)
2	6	Animal died under anesthesia
3	28	Induced pre strain during surgery resulted in out-of-range strain recording. Device was recovered for analysis
4	21	Device re-positioned and device body folded on itself while implanted resulting in diminished RF performance. Device was recovered for analysis
5	22	Encapsulation failure
6	43	High stresses in the tabs holding the device body showed wear and cracking in Parylene encapsulation resulting in circuit damage.
7	5	Animal death due to undetermined cause.
8	25	Defect in encapsulation on self-similar serpentine.
9	Alive	Animal still alive
10	27	Device recovered for SEM bone attachment analysis.

6. Discussion (page 16). A limitation of current study is that the device was not entirely on the bone surface (due to inherent size constraints) in the chronic rat study (Fig. 4A). What aspects still need to be evaluated in large animals studies should be mentioned here.

Our response: We agree with the reviewer that investigations in large animal models are required to unlock the full potential of osseosurface electronics. The larger available footprint on the bone of large animals enable more elaborate sensors placement and multimodality to investigate bone health, for example with photometric sensors. Additionally, chronic sensor readout is a challenge if there is no infrastructure with high powered readers or small readers in direct proximity are present^{R2}. Luckily there are several great approaches in current literature that are available that can solve some of these challenges in future work. In small animal models we believe that the current device size is not so much of a limitation as animals tolerate devices well and no measurable effect on behavior is observable. We have added these considerations to the discussion section.

R2. Lin, R. *et al.* Wireless battery-free body sensor networks using near-field-enabled clothing. *Nat. Commun.* **11**, 1–10 (2020).

Modification made to the manuscript (page 23, line 386):

“These results demonstrate the feasibility of operating osseosurface electronics with a smartphone in an at-home setting. Locations covered with several muscle layers such as the femur in larger animals, can be accessed with a dedicated reader solution with near-field-enabled clothing for efficient power transfer and continuous communication during everyday behaviours⁴⁶. Figure 67F plots the relative data rate and harvested power as functions of tissue thickness between the reader and the device, showing no degradation in either data rate or available power level at the devices at tissue thickness up to 11.5 cm.”

Additions made to References (Ref 46):

46. Lin, R. et al. Wireless battery-free body sensor networks using near-field-enabled clothing. Nat. Commun. 11, 1–10 (2020).

7. Methods. Details for the computational electromagnetic design should be provided.

Our response: We thank the reviewer for pointing this out. In this work we have used experimental determination of antenna designs that are based on simulations conducted in previous work. We have clarified this context in the methods section.

Modification made to the manuscript (page 26, line 456):

“The completed wireless power harvesting module was tuned to 13.56 MHz using a spectrum analyzer with reflection bridge (SSA 3032X, Siglent). Capacitors were added to tune the antenna to reach peak attenuation at 13.56 Mhz. The capacitors are used to calculate inductance of the antenna by matching the reactance of the capacitor and

inductor at 13.56 Mhz. Power and voltage characterization of the antenna was tested by placing the device in the ~~and placed at the~~ center of the test arena in parallel with the arena floor. The rectified voltage was then recorded with a digital multimeter while the electrical load was varied from $\sim 50\Omega$ to $\sim 5\text{ k}\Omega$.”

8. Fig. S5. Inductance and resistance of the antenna should be reported.

Our response: We thank the reviewer for this response and have corrected the manuscript to describe the process of antenna optimization in both the results and methods sections.

Modification made to the manuscript (page 7, line 134):

“Both scenarios required robust power transfer and efficient wireless power transfer via magnetic resonant coupling that can be boosted by adopting an ~~RX antenna secondary antenna~~ with high quality factor (~~Q-factor~~), in our case antennas with high inductance and low impedance¹⁸. Figure 2A displays a large animal model device with ~~optimization though iterative imperial electromagnetic designs (Fig. S6) of the secondary antenna using $\sim 185\text{ pF}$ to match the reactance of the antenna inductance of $\sim 745\text{ nH}$ (3 turns, $600\text{ }\mu\text{m}$ wide, $60\text{ }\mu\text{m}$ spacing) at 13.56 MHz allowing for low trace impedance ($\sim 30\text{ m}\Omega$)¹⁹, for operational voltages of 2.5 V ; ~~a limitation introduced by the CMOS technology used by the NFC SoC (details in Supplementary Fig. S4-5)~~. As shown by the power harvesting characteristics (Fig. 2B and Fig. S7) measured at the center of a handheld ~~primary TX~~ antenna (diameter $\sim 20\text{ cm}$), the maximum values of harvested power ($\sim 14\text{ mW}$) and rectified voltage ($\sim 2.1\text{ V}$) support operation at an electrical load of $\sim 300\text{ }\Omega$.”~~

Additions made to References (Ref 34, Ref 35):

19. Jiang, C., Chau, K. T., Liu, C. & Lee, C. H. T. An Overview of Resonant Circuits for Wireless Power Transfer. *Energies* vol. 10 (2017).

Modification made to the manuscript (page 8, line 151):

“To provide power to freely moving small animals, critical for exploratory research in large test arenas (treadmill cage, $45\text{ cm} \times 12\text{ cm}$; home cage, $26\text{ cm} \times 33\text{ cm}$), devices designed for rat models (Fig. 2D) require an enlarged ~~RX antenna secondary antenna~~ ($3.5\text{ cm} \times 2.5\text{ cm}$) and device layout that features serpentine interconnects ($\sim 11\text{ cm}$ at full extension) to route the biointerface from the back of the subjects which houses the electronics and antenna section of the device to the limb that is the sensing target in our experiments. ~~Optimization though iterative imperial electromagnetic designs Computational-electromagnetic designs (Fig. S6 and Fig. S7) of the secondary antenna using $\sim 485\text{ pF}$ to match the reactance of the antenna inductance of $\sim 284\text{ nH}$ (2 turns, $600\text{ }\mu\text{m}$ wide, $60\text{ }\mu\text{m}$ spacing) at 13.56 MHz allowing for low trace impedance ($\sim 3.3\text{ m}\Omega$)¹⁹, enables harvesting performance of a rectified voltage of $\sim 2.2\text{ V}$ at a load of $300\text{ }\Omega$ (16.13 mW) at the center of the $45\text{ cm} \times 12\text{ cm}$ cage, providing a margin of 0.4 V to enable constant system voltage of 1.8 V throughout all experimental conditions.”~~

Additions made to References (Ref 10):

19. Jiang, C., Chau, K. T., Liu, C. & Lee, C. H. T. An Overview of Resonant Circuits for Wireless Power Transfer. *Energies* vol. 10 (2017).

9. Fig. S14. What specific tissue region around the device is the histology from?

Our response: We thank the reviewer for pointing out this missing information and have specified location of the histology in the figure caption and the methods section and have included new images of device explantation in the supplementary information.

Modification made to the manuscript (page 19, line 319):

“The welfare of the test animals is also reflected by the steady weight-gain from two days post-surgery until the study is terminated (Fig. S20~~Fig. S13~~). In addition, histological analysis shows that the **subcutaneous implantation body of the device body in the lumbar vertebra region**, is surrounded with fibrous tissue containing blood vessels without any evidence of inflammation or significant foreign body response after two weeks of in-vivo operation (Fig. S21A-E).”

Modification made to the manuscript (page 33, line 606):

“After 2 weeks the rats were euthanized. Following euthanasia device placement was characterized by scanning rats at 20 μm resolution using a Siemens Inveon micro-CT Scanner. For histology, tissue surrounding the implanted device was excised **and imaged using an optical microscope (Wild M3Z Stereozoom Microscope, Leica) with an attached camera (iPhone 12 Pro, Apple) (Fig. S21A-C)**, fixed in 10% formaldehyde for 24 hours and embedded in paraffin. Ten-micron sections were cut and stained with hematoxylin and eosin (Fig. S23D-F). Slides were viewed using a Nikon microscope coupled to a Nikon DS-Fi1 camera at magnifications ranging from 20x – 100x.”

Addition to Supplementary Information (Figure S21):

Figure S21. Tissue analysis surrounding the device after explantation. A, Photograph of tissue growth around device. **B,** Photograph of tissue under microscope. **C,** Photograph of strain gauge glued to the bone. **D,** Histology of tissue sample surrounding device with 20x magnification. **E,** Histology of tissue sample surrounding device with 40x magnification. **F,** Histology of tissue sample surrounding device with 100x magnification.

Reviewer #3 (Remarks to the Author):

General comments:

Le Cai and co-workers here report on a type of flexible, wirelessly powered electronic biointerface patch that can be attached to bone surfaces for chronic implantation, that the authors consequently term "Osseosurface electronics".

The paper is well organized and written, and certainly adds an important piece to the growing field of bioelectronics. However, the device architecture, methods of fabrication, wireless data readout and set of sensors (strain, temperature, and a u-ILED) closely resemble those of previous work from the group and others in the field. While this should not per se be understood as criticism on the work itself, a few points should be made clearer in order for this platform to become more convincing.

Our response: We thank the reviewer for the positive input and constructive suggestions. Based on the reviewers comments we have performed additional experiments and have revised the manuscript. Based on these changes we think we have added substantial impact specifically outlining unique technologies enabling chronic osseosurface electronics and clarified questions of the reviewer.

Specific points:

As I understood, the main idea of mounting this type of electronics on bone is to increase chronic operational lifetime by reducing mechanical impact associated with animal (and later eventually human) motion on the electronics itself. On page 17 the authors however state that their current method of bonding using cyanoacrylates limits lifetime to weeks ("Currently we expect device lifetimes on the order of several weeks limited by the cyanoacrylate attachment to the bone, an interface that gradually degrade"), taking into account only said interface though not the operational lifetime of the electronics under in vivo conditions itself. In the current manuscript, experimentally supported arguments on this form of electronics being more chronically stable than similar, previously reported forms are moderately convincing for the following points:

Our response: We agree with the reviewer that one of the many Osseosurface electronics benefits are chronic operation on bone. There are also several additional benefits such as new biointerface categories that enable new experiments to evaluate bone health. In order to make this manuscript more impactful we have conducted several new experiments that demonstrate chronic operation and also demonstrate techniques that enable the ability to grow devices to the bone to enable permanently bonded osseosurface electronics. We believe characterizing these capabilities serves to establish this bioelectronic mounting location as a platform for sensors to assess bone health, surrounding tissue health and as a mounting location for chronic device implantation to enable bioelectronic medicine.

1) Many of the bench top cyclic loading experiments are performed for a very limited number of cycles only, a few tens of seconds of operation are shown mostly (i.e. Figure S8 for tests of the bone mounted strain sensors, Figure 3 B,C,H only a

few cycles are shown). Maybe i missed it somehow, but it seems especially such bench top experiments with the devices mounted on bones or bone models lend themselves to corroborate long term (mechanical) stability of the devices on such interfaces.

Our response: We thank the reviewer for pointing this out and we have added several bench top experiments to demonstrate chronic stability of the devices. This includes accelerated rate tests and strain experiments to showcase interface and device stability. We have also included new animal experiments that show device operation over several weeks.

Modification made to the manuscript (page 15, line 248):

“The durability of the serpentine interconnect is critically important for reliable in-vivo operation. FEA simulation guided design enables reliable performance under repeated strains of over 250% while maintaining a maximum strain of ~ 0.7% in the ~~Cu~~copper traces which is below 1%, the failure strain of ~~Cu~~copper(Fig. 4B)³⁵. Chronic ~~electro-mechanical~~ stability is confirmed by cyclic straining to 250% for 10,000 cycles, with negligible change in conductivity (Fig. 4C). ~~Critical to the stability of sensor readings during behavior is the mechanical isolation of the stretchable interconnect and the strain sensor, this includes introduction of strain to the bone during deformation of the serpentine interconnect. The biointerface is mechanically isolated by the serpentine interconnects designed to transfer minimal strain during deformation to the bone and the strain sensor biointerface, as confirmed by FEA in Fig. 4D (Fig. S15) showing that stretching the serpentine by 250% does not influence the sensitivity or accuracy of the strain gauge (Fig. 4E-F and Fig. S16, details in Method section),~~which is further. Further evaluation ~~validated~~of serpentine electro-mechanics are performed with ~~by~~ bench-top experiments using a ~~servo-hydraulic materials testing system (MTS) indicating stable operations of recording capabilities under multiple cycles of high strains tests~~(Fig. S17) above physiological strains as means to evaluate robustness of the approach. Both benchtop electro-mechanical testing of the monolithic serpentine structure with FEA simulations of the serpentine shows solid performance in strain isolation enabling flexible sensor placement to distal regions with minimal effect of sensor readouts.

In-vivo studies in rodents. *The subdermally implantable form factor and wireless, battery-free operation enable an in-vivo multimodal bi-directional interface with the musculoskeletal system without compromising the free motion of subjects in various test arenas (Fig. 5).”*

Addition to Supplementary Information (Figure S17):

Figure S17. Extended servo-hydraulic materials testing. A, 10,000 cycle testing on a MTS (Series 810, MTS Systems Corporation). B, First 10 cycles of testing procedure. C, Last 10 cycles of testing.

2) There are tests where the devices are submerged in PBS for about 40 h, and continue to transmit data wirelessly. I wonder why only readings of the temperature sensor are shown, but not the ones from strain sensors and eventually a test including the u-ILEDs? I am aware that such tests of device stability are challenging, but a clear advantage of the integration on the bone surface in terms of device stability should be demonstrated in such experiments.

Our response: We thank the reviewer for the suggestions and we have prepared a range of accelerated rate tests to characterize device stability while monitoring temperature recording capability, digital system function, RF properties and LED function. We have added results to the manuscript and conclude a theoretical lifetime of years with current encapsulation schemes.

Addition to Supplementary Information (Figure S11):

Figure S11. Accelerated rate testing at 90 °C, 60 °C, and 40 °C in PBS, testing for wireless communication function, optical stimulation function, standing wave ratio of antenna (Red, Orange, Yellow), and wireless recording of temperature (Navy, Light blue, Blue).

Modification made to the manuscript (page 11, line 187):

“Long-term data recording (Fig. 2I) tests performed in the biologically relevant settings with a moving osseosurface electronic devices, (details in Methods section) results in stable communication over extended periods of time (42 hours, no limitation in operation time) where the recorded temperature profile matches that of environment temperature measured by a spatially separate thermometer. Further studies of chronic device encapsulation (18 μm Parylene-C) and device operation are conducted via an accelerated rate test in 90 °C, 60 °C and 40 °C PBS bath to enable device lifetime estimation (205 days at 37 °C) via Arrhenius scaling²¹ (Fig S11).

Biointerface characterization. Characterization of the multimodal biointerface is performed with benchtop experiments that reflect the in vivo environment.”

Additions made to References (Ref 21):

21. Shin, G. et al. Flexible Near-Field Wireless Optoelectronics as Subdermal Implants for Broad Applications in Optogenetics. *Neuron* **93**, 509-521.e3 (2017).

Modification made to the manuscript (page 27, line 476):

“Long-term data recording was performed using the rodent device and the 45 cm x 12 cm ~~TX antenna~~ primary antenna over a ~ 42-hour period while the device was immersed in PBS solution and constantly moving back-and-forth (25 cm/s) on a custom-built rat treadmill. A custom-built thermometer (LMT70, Texas Instruments) was used to monitor the environment temperature. Device lifetime estimation was calculated using an activation energy of 57800 J/mol based on device failure at 90 °C and latest measured operation of device at 60 °C. Using the Arrhenius scaling equation, we estimated a device lifetime of 205 days at 37°C⁴⁹.

Characterization of the wireless strain sensor: Bench-top tests of the wireless strain sensor were performed using an explanted sheep’s femur.”

Additions made to References (Ref 49):

49. Menon, P. R., Li, W., Tooker, A. & Tai, Y. C. Characterization of water vapor permeation through thin film Parylene C. in *TRANSDUCERS 2009 - 2009 International Solid-State Sensors, Actuators and Microsystems Conference 1892–1895 (2009)*. doi:10.1109/SENSOR.2009.5285687.

3) Limiting factors on in-vivo lifetime of the devices beyond a failure of the cyanoacrylate bonding should be discussed and eventually evaluated in bench-top experiments. I would assume that issues on bodily fluids penetrating encapsulation layers should arise. Here, it would be interesting to see if the bone-electronics interface would actually have some additional benefits in terms of reduced exposure to body fluids, increasing lifetime? Given the difficulty and time and cost expense of in-vivo studies, i would suggest a more thorough bench top characterization similar to the one demonstrated in Figure 2I, but including all sensor/actuator components of the electronic device.

Our response: We thank the reviewer for the constructive suggestions and have demonstrated what we alluded to in the discussion of the original manuscript. We have used CPC particles as an agent to grow devices to the bone to enable permanent bonding. In addition, we have shown real time recording on freely moving animals and long-term operation in subjects. Corresponding changes to the manuscript are outlined in answer to reviewers 1 and 2 and changes that document CPC particle aided sensor growth are outlined below.

Modification made to the manuscript (page 15, line 247):

*“In addition, histological analysis shows that the **subcutaneous** implantation **body of the device body in the lumbar vertebra region**, is surrounded with fibrous tissue containing blood vessels without any evidence of inflammation or significant foreign body response after two weeks of in-vivo operation (Fig. S23A-E).*

Permanently attached Osseosurface electronics. *The ultrathin electronic platform enables direct lamination onto the bone which provides the opportunity to attach devices permanently to gather information on bone health long term. Because of cell turnover any glue will exhibit limited lifetime and continuously degrading biointerface quality. A solution to this problem is to directly grow osseosurface devices to the bone using of CPC particles³⁶. For the use in animal subjects fast adhesion is critical to enable accelerated experimental timelines, which can be accomplished by the addition of transforming growth factor beta 1 (TGF- β 1), an osteogenic protein that enables rapid bone bonding to the CPC particles³⁷. We use this scheme to bond CPC particles to the osseosurface electronics using implant grade epoxy, and subsequently applying TGF- β 1 to the CPC particles. The resulting layered device composition is shown in Fig.6A. The exposed particles enable effective bone bonding after implantation and temporary fixation with resorbable sutures (Fig.6B). The efficacy of this approach can be observed in vivo by recording the average delta strain values of gait on a treadmill. Figure 6C shows the evolution of the delta strain over the course of several weeks, here we can observe weak bonding of the strain gauge to the bone for the day of the surgery with rapid increase in delta strain values in the following days and a stable attachment after a week (Fig. S15). The successful growth of*

the device to the bone can also be observed after explanation (day 27) and cross-section preparation and subsequent secondary electron imaging (Fig.6D and Fig. S16), details in methods section. The micrograph shows successful growth of the bone to the particles that permanently affix the flexible electronics enabling chronic recording of bone health. The chronic application of these devices enables a multitude of new studies for example strain mediated bone remodeling with experimental paradigms that involve extensive mobility of subjects and longitudinal studies with multiple co-habiting subjects which is facilitated by complete recovery after implantation of the wireless and battery free device as shown in Fig.6E.”

Additions made to References (Ref 36):

36. Szivek, J. A., Anderson, P. L., Dishongh, T. J. & DeYoung, D. W. Evaluation of factors affecting bonding rate of calcium phosphate ceramic coatings for in vivo strain gauge attachment. *J. Biomed. Mater. Res.* 33, 121–132 (1996).
37. Cordaro, N. M., Szivek, J. A. & DeYoung, D. W. Surface enhancements accelerate bone bonding to CPC-coated strain gauges. *J. Biomed. Mater. Res.* 56, 109–119 (2001).

Modification made to the manuscript (Page 26, Line 480):

"Finally, the applicator was carefully peeled off with minimal out-of-plane shear force between the membrane.

Calcium phosphate ceramic coating

Implant-grade epoxy (Master Bond EP42HT, Master Bond, Inc., Hackensack, NJ) was prepared according to the manufacturer's instructions and a thin even layer was blade coated to the bottom surface of the strain gauge (N2A-06-S5182N-10C/E4, Micro-Measurements). Spherical crystalline calcium phosphate ceramic (CPC2, CeraMed) was added to cover the surface of the epoxy and silicone rubber was used to gently apply downwards pressure on the CPC particles while curing the epoxy for 24 hours at room temperature. All CPC coatings were inspected for surface exposure without epoxy coating to enable bond to the bone. Devices were coated with TGF- β 1 to accelerate bone to CPC bonding.

Scanning electron microscope imaging of CPC – bone adhesion

A rat after 27 days post-implantation was euthanized, and the femur was explanted along with the attached strain gauge. The bone was cleaned, removing any soft tissue and fixed using a neutral buffered formalin for 2 hours. The bone was dehydrated in a solution of ethanol (EtOH) starting at 70% for 2 hours and increasing concentrations for at least 2 hours as follows: 80% EtOH, 95% EtOH, 95% EtOH, 100% EtOH, 100% EtOH, xylenes, 100% EtOH. The bone was embedded in methyl methacrylate (Technovit 9100 kit, Kulzer) through a pre-infiltration process for 2 hours at room temperature and an infiltration process within a vacuum chamber for 15 minutes to remove any bubbles. The embedded bone is then cured at -4C over night. The bone and strain gauge were cut exposing the cross-sectional view. The cross-section surface was smoothed and polished using a grinding wheel (GP-25, Leco) from 120 grit to 600 grit. The surface was gold sputtered (Hummer 6.3, Anatech USA) allowing for a gold thickness of 9 nm. SEM images were taken with various magnifications (60x, 120x, 280x, 300x, 600x) around the bone strain gauge interface using (Inspect S50, FEI) operating at 30kV.

Modification made to the Figure (Figure 6):

Figure 6. Promotion of osteogenesis using surface-engineered calcium phosphate ceramic coatings. A. Photograph of the bottom coated with CPC and a side view diagram of layer composition of the strain gauge sensor with CPC. **B.** Image of device 3 weeks post implantation. **C.** Wireless strain gauge data collected over 2 weeks post implantation showing stable CPC adhesion. **D.** Single electron microscope image of cross section of CPC adhesion to bone. **E.** Animal healing after implantation.

Addition of Supplementary Information (Fig. S23):

Figure S23. Additional SEM cross section images of CPC bone bonding. A. SEM cross section of strain gauge bonded to the bone using CPC particles with a magnification

of 120x and a photograph image inset. **B.** SEM cross section with a magnification of 280x, **C.** 300x, and **D.** 600x.

Modification made to the manuscript (page 24, line 399):

"This offers unprecedented opportunities for mechanistic studies of osteogenesis and pathogenesis of musculoskeletal diseases, as well as the development of new types of diagnostics and therapeutics. Furthermore we demonstrate the successful attachment of the Osseosurface electronic biointerface to the bone utilizing surface engineered CPC particles, an important aspect of this technology that enables recording on the order of many years enabling device operation for the lifetime of the subjects without requiring secondary surgery. ~~Currently we expect device lifetimes on the order of several weeks limited by the cyanoacrylate attachment to the bone, an interface that gradually degrade 41,42. Future development can enable the formation of a permanent interface with the bone by exploring bioactive coatings that facilitate direct integration with the osseosurface, allowing device operation for the lifetime of the subject without requiring secondary surgery.~~

This chronic platform can then enable acquisition of holistic data of bone health status and closed loop therapeutic intervention to facilitate treatment and rehabilitation."

Modification made to the abstract (page 2, line 36):

"Battery-free device architecture, direct growth to the bone via surface engineered calcium phosphate ceramic (CPC) particles, demonstration of operation in deep tissue in large animal models and readout with a smartphone highlight suitable characteristics for exploratory research and utility as a diagnostic and therapeutic platform."

Modification made to the abstract (page 32, line 588):

The strain gauge was fixed to the femur using a cyanoacrylate adhesive (M-Bond 200, Micro-Measurements). ~~For implantation of CPC coated devices in small animals, a 5-0 vicryl suture lassoed around the biosensor was used to pass the sensor from the lumbar region through the subcutaneous tunnel to the thigh. This lasso technique was used to avoid manipulation of the device with sharp surgical instruments and minimize the risk of damage to the encapsulating layers. CPC coated gauges were secured to bone using two 5-0 vicryl sutures without adhesive.~~ Layered closure was performed using 5-0 vicryl for fascia and 4-0 Quill for subcutaneous tissues prior to recovery from anesthesia.

Modification made to the manuscript (page 34, line 640):

"Adhesive agent (M-Bond 200, Micro-Measurements) was then applied to the bottom side of the device. The applicator was designed to make surgical process in large animals easier by improving ability to conformally mount the strain gauge to the bone in one step. ~~The applicator was subsequently placed on the bone and pressed gently against the bone in order to conform the device to the bone surface. Next, Once placed, the device was then firmly pressed against the bone with an even force until the adhesive agent cured.~~ Finally, the applicator was carefully peeled off with minimal out-of-plane shear force between the membrane, device encapsulation, and adhesive agent."

4) The rodent study summarized in Figure 4 demonstrates that implantation of the devices has little impact on the gait of the rodents for at least two weeks. But it seems it does not demonstrate device functionality beyond 10h (Figure 5B), or did i miss this? Again, the two in vivo recorded strain cycles of Figure 5 C make me wonder if the authors could include more of their data to underline functionality of the implanted platform.

Our response: We thank the reviewer for the suggestions and have added real-time data acquisition of subjects on a treadmill and have added recordings from animals with devices that where implanted for several weeks.

Modification made to the manuscript (page 18, line 299):

“Strain profile recorded wirelessly in a custom treadmill arena with the 45 cm x 12 cm ~~TX antenna~~ primary antenna reveals characteristic loading and unloading phases with absolute strain values comparable to literature values for the rat femur obtained by tethered sensors²².

Time synced wireless recording of strain and temperature are compared with video recording of the animal during walking periods after a week of recovery (Supplementary Video V2). Analysis enables intimate insight into bone strain of the femur during gait. The recordings of strain show four distinct points during a gate cycle: 1) Mid-stance, 2) Lift-off, 3) Mid-swing, 4) Touch-down^{37,38}. These points and their corresponding frames are shown in Figure 5C. Behavioral assays, gait analysis in particular, are useful tools to investigate pathogenesis and develop new therapeutics for musculoskeletal disorders such as osteoarthritis³⁹.”

Additions made to References (Ref 34, Ref 35):

37. Fujiki, S. et al. Adaptive hindlimb split-belt treadmill walking in rats by controlling basic muscle activation patterns via phase resetting. Sci. Rep. 8, 17341 (2018).

38. Dienes, J. A. et al. Analysis and Modeling of Rat Gait Biomechanical Deficits in Response to Volumetric Muscle Loss Injury. Front. Bioeng. Biotechnol. 7, 146 (2019).

Modification made to the Figure (Figure 5):

Figure 5. In-vivo studies of OSE in rats. **A.** Photograph of the rear part of a rat featuring the μ -ILED attached on the femur illuminating through muscle and skin. **B.** Temperature profile recorded in-vivo during the period of ~ 11.5 hours following the implantation surgery. Inset, schematic of the rat residing in the home cage. **C** Time synchronized strain recording and corresponding video frames of the rat gait cycle. *Strain profile recorded in-vivo while the rat's rear leg is being pulled manually.* **D-E.** Comparison of the rat's gaits at different stages of the study: before surgery, 1 week after surgery and 2 weeks after surgery: photograph of rats walking on the treadmill overlaid with the trajectory of the ankle (**D**); key parameters – stride frequency, stride length and duty factor – that characterizes the rat's gaits (**E**). Color of the dots in **D** represents the velocity of the ankle.

Addition made to the Supplementary Information (Video V2):

Video V2. Video with corresponding plot of wireless recording of strain of the left femur during gait.

5) Figure 4, the rodent study. I may have missed it, but the u-ILED seems not be involved in the recording of any physiological data or any form of (measured) optical stimulation. If the authors choose to include this part in their Figure, i would recommend to at least demonstrate that the u-ILED remains functional for the period of implantation, opening up the possibility for further studies involving this type of devices for optical bio interfaces.

Our response: We thank the reviewer for the constructive suggestions and have added operation of the LED in freely moving subjects and have added additional supplemental figures to document device operation.

Modification made to the manuscript (page 17, line 281):

“The optoelectronic interface attached on the bone surface allows for direct optical stimulation of skeletal muscles in freely-moving subjects, as shown in Fig. 5A where illumination of the muscle is visually validated, highlighting a systematic advantage over traditional transdermal light sources³⁰ or optical fiber-based techniques²⁹ which are challenging to implement in highly mobile areas. Light delivery can be programmed to control frequency and duty cycle of stimulation each with 16-bit precision. Chronic functionality is demonstrated with in frame analysis of tracked red pixel intensity over time (Fig. S19) from video recordings 22 days post implantation (Supplementary Video V1). In-vivo device operation of temperature sensing capabilities is tested over extended time periods ~~is~~ demonstrated by a 10-minute moving window of thermographic recording that show recovery of the animal post-surgery for over 11 hours in the home cage (Blue) with a standard deviation within the 10-minute moving window (pink) (Fig. 5B).”

Addition to Supplementary Information (Figure S19):

Figure S19. Tracked red light intensity of optogenetic stimulation while implanted with control of frequency and duty cycle: 9 Hz, 50% (red); 9 Hz, 100% (green); Off (yellow); 5 Hz, 75% (blue).

Addition to Supplementary Information (Video V1):

Video V1. Video of light delivery with biointerface implantation location on the rat femur.

In summary, the authors present a very interesting electronic platform that interfaces with bone surfaces. However, the current in vivo demonstrations in rodents and ex-vivo experiments with sheep bones or on deceased sheep seem to be not ideal in transporting the idea of having developed a chronically stable biointerface. Here, functionality of the whole multimodal devices is not demonstrated very convincingly. I recommend to mitigate those issues mostly via bench-top experiments involving explanted bones or bone surrogates and PBS. Demonstrating a clear advantage of the bone-electronics interface on device longevity would certainly multiply the impact of this study.

Our response: We thank the reviewer for the positive inputs and all constructive suggestions and hope we could improve the manuscript substantially based on the new experiments outlined in this response

REVIEWERS' COMMENTS

Reviewer #1 (Remarks to the Author):

The authors have done a very impressive job in the revision. All my previous concerns have been addressed. The publication is recommended.

Reviewer #2 (Remarks to the Author):

The authors have satisfactorily addressed all of my comments. I appreciate their substantial efforts in making the revisions.

Reviewer #3 (Remarks to the Author):

The authors have extensively revised their manuscript, added important new experimental evidence and clarifications to address most of the reviewer's comments quite adequately. I would now certainly recommend the manuscript for publication.